# MicroRNA expression profile in *Lampetra morii* upon *Vibrio anguillarum* infection and miR-4561 characterization targeting *lip*

Lie Ma[1,2,3], Meng Gou[1,2,3], Zeyu Du[1,2,3], Ting Zhu[1,2,3], Jun Li[1,2], Qing Wei Li[1,2] & Yue Pang [1,2✉]

As a critical evolutionary pivot between invertebrates and vertebrates, lampreys provide rich genetic information. Lamprey immune protein (LIP) is a key immune regulator. MicroRNAs, well-conserved in the response to immunological stress, remain understudied in lamprey immunity. We generated a lamprey microRNA expression atlas, using deep sequencing, upon *Vibrio anguillarum* infection. Using comparative methods, we found that miR-4561 potentially regulates innate immunity via interaction with *lip*. We found a sequence in the 3′-UTR region of LIP mRNA complementary to the miR-4561 seed region; miR-4561 expression was negatively correlated with LIP. During *V. anguillarum* infection, miR-4561 inhibited LIP expression and bacterial clearance. Notably, LIP expression in supraneural body cells was necessary for the Gram-negative immune response. Additionally, we observed that over-expression of miR-4561 induced apoptosis in embryonic cells, suggesting a role in embryonic development. Collectively, we show lamprey microRNAs may significantly affect gene regulation and provide new insights on LIP-mediated immune regulation.

[1] College of Life Science, Liaoning Normal University, Dalian, China. [2] Lamprey Research Center, Liaoning Normal University, Dalian, China. [3] These authors contributed equally: Lie Ma, Meng Gou, Zeyu Du, Ting Zhu. ✉email: pangyue01@163.com

Lamprey, the jawless vertebrate, belongs to the superclass Cyclostomata and represents the oldest group of vertebrates[1,2] Lamprey provides rich genetic information for the origin and evolution of vertebrate, and is known for its unique adaptive immune system based on variable lymphocyte receptor (VLR) antigen recognition. Unlike IgG or IgM in mammals, the variable lymphocyte receptor of the lamprey consists of a variable leucine-rich repeat sequence, which provides high value for immunological research in lamprey[3–5]. Therefore, the lamprey is considered a model animal to study vertebrate evolution, development, and origin of immune system[6].

MicroRNAs (miRNAs) are small non-coding RNAs with a length of 19–23 nucleotides (nt) that, among other mechanisms, regulate mRNA expression by binding to the 3′ untranslated region (3′-UTR) of the target mRNA[7–9]. Moreover, miRNAs can have post-transcriptional regulatory functions, including mRNA degradation and mRNA translational inhibition. Owing to their broad target spectrum, miRNAs are implicated in many physiological and pathological responses, including host defense regulation and the inflammatory response[10]. Indeed, many miRNAs involved in a plethora of biological processes, such as development, metabolism, immunity, and reproduction, have been found in aquatic vertebrates such as fish[11–14]. In a 2010 study, the authors generated small RNA libraries from ammocoete larvae of the brook lamprey (Lampetra planeri) and nine tissues (brain, gills, gut, heart, kidney, liver, mouth, muscle, and skin) of a single sea lamprey adult (Petromyzon marinus)[15]. They found that miRNA expression patterns were conserved and comparable to miRNA found in hagfish and gnathostomes. Their results imply that the organ-specific expression of miRNA is evolutionarily conserved and their functions are consistent with their appearance in the genome. Thus, the high degree of evolutionary conservation of miRNA families in vertebrates supports the view that miRNAs play a key role in vertebrate gene regulation. However, the involvement of miRNAs in the antibacterial defense in the lamprey has not been investigated yet.

In our previous studies, we identified a protein called lamprey immune protein (LIP) involved in lamprey innate immunity. LIP is expressed in the supraneural body tissue and is characterized by an efficient and selective cytotoxic activity against various tumor cells, through the dual selective recognition and efficient binding to two N-linked glycans on GPI-anchored proteins (GPI-APs) and sphingomyelin (SM) in lipid rafts[16,17]. LIP is a 313-amino-acid protein composed of an N-terminal jacalin-like domain and a C-terminal aerolysin domain. Although in our previous study we suggested that LIP can up-regulate the expression of caspase 1, RIPK1, and RIP3 to trigger pyroptosis and necroptosis, hypothesizing that LIP is an immune factor involved in host immunity[16], a complete description of LIP functions and regulation is still lacking. Moreover, the regulatory mechanism of LIP in lamprey immunity and development needs to be explored.

In this study, we investigated the miRNA expression profiles in the leukocytes of Lampetra morii under physiological conditions and upon infection by the Gram-negative bacterium Vibrio anguillarum, assessed at two different time points. The aim of the study was to perform a comparative analysis of miRNA expression during bacterial infection and identify potential regulators of the immune response.

## Results

### Small RNA sequencing and miRNA identification.
To investigate the miRNA expression profiles in leukocytes of the lamprey Lampetra morii before and after the infection with V. anguillarum, we generated three small RNA libraries obtained from normal lampreys and V. anguillarum-infected lampreys collected after 8 hours (h) and 17 days (d) of infection. We then sequenced the libraries using the Illumina deep-sequencing platform. We generated 15,282,313, 16,666,925, and 12,082,914 raw reads from libraries obtained at 0 h (uninfected), 8 h, and 17 d, respectively. After filtering out low-quality sequences, adapter contaminants, and poly-A sequences, we obtained the following reads: 15,000,950 (98.16% of the raw reads), 16,120,703 (96.72% of the raw reads), and 11,192,192 (92.63% of the raw reads) for the 0 h, 8 h, and 17 d libraries, respectively.

Sequence length distribution showed a wide variation, ranging from 18 to 35 nt. As expected, we observed an enrichment for 22, 23, and 24 nt small RNAs in all libraries, which is compatible with the presence of miRNA (Fig. 1a). We then annotated the following non-coding RNAs, using the Rfam and GenBank databases: rRNA, tRNA, snRNA, snoRNA, known_miRNA, novel_miRNA, and repeated sequences (Fig. 1b). We identified 92 pre-miRNA, of which 28 were known and 64 were novel, encoding 21 known and 61 novel mature miRNAs, respectively (Supplementary Data 1). Sequencing data have been submitted to the NCBI Sequence Read Archive (SRA) under the following accession numbers: SAMN14377427-SAMN14377429.

### Identification of differentially expressed miRNA upon bacterial infection.
We then determined the differentially expressed miRNA following bacterial infection and performed cluster analysis, using a hierarchical clustering method based on the relative expression level of miRNA expressed as log2 (ratios). We identified 7 down-regulated miRNAs (miR-4543, miR-novel-8, miR-4561, miR-23b, miR-novel-2, miR-144-5p, and miR-497-3p) and 6 up-regulated miRNAs (miR-level-94, miR-level-73, miR-level-77, miR-level-1, miR-29a-5p, and miR-451) 8 h after the infection. Similar expression patterns of miRNAs imply similar functions. Thus, we then performed cluster analysis of the miRNA expression pattern by pair-wise comparison among the three samples (8 h vs 0 h, 17 d vs 0 h, and 17 d vs 8 h) (Fig. 1c). We found 60 miRNAs, of which 13 were up-regulated or down-regulated between 0 h and 8 h; 24 were down-regulated between 0 h and 17 d, and 23 were up- or down-regulated between 8 h and 17 d (Fig. 1d). The expression level of these miRNAs displayed a high degree of variation, ranging from 1 to 1,356,331 (Tables S2–S4). We identified two miRNAs, miR-4561 and miR-novel-14, which were differentially expressed among all groups. After performing a bioinformatics analysis to predict miRNA targets based on sequence complementarity, sequence context, and conservation across species, we found that, intriguingly, miR-4561 (whose secondary structure is shown in Fig. 2a) was the miRNA with the highest score for the lip gene (Table S5). Indeed, LIP mRNA contained the seed sequence for miR-4561 in its 3′-UTR. (Fig. 2b, c). On the lamprey genome, the pre-miR-4561 is located between the fifth and sixth exons of the rps3 gene (GL486431:6793-6871) (Fig. 2d).

### Expression level of miR-4561 and lip are affected by bacterial infection.
We investigated the expression pattern of miR-4561 in various tissues of the adult lamprey by quantitative RT-PCR. We found that miR-4561 was constitutively expressed in the heart, brain, and supraneural body (Fig. 3a). After infection by V. anguillarum or stimulation by LPS, a component of the Gram-negative bacterial cell envelope that triggers the inflammatory response, the expression level of miR-4561 decreased significantly after 2 h, and began to recover after 8 h. At 72 h, the expression level recovered significantly but was still lower than the pre-experimental expression levels. Interestingly, the lip expression profile had the opposite trend, increasing between 2 and 24 h, and decreasing from 24 to 72 h (Fig. 3b, c). We also detected the expression of LIP protein by western blotting. Results showed

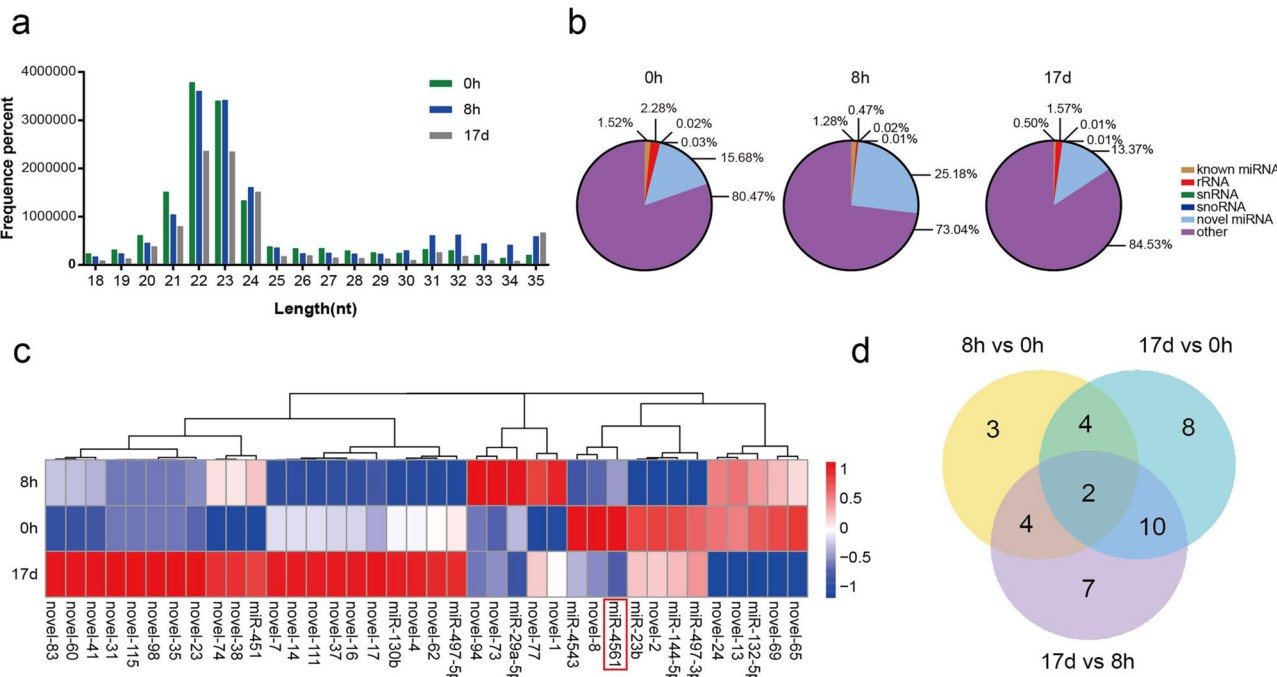

**Fig. 1 Illumina sequencing analysis of miRNA expression in lamprey following bacterial infection. a** Length distribution of the RNA reads in the three libraries (0 h, 8 h, and 17 d). **b** Small RNA type composition. **c** Cluster map of miRNA differentially expressed in the three groups. **d** Venn diagram of differentially expressed miRNAs in the following comparisons: 8 h vs 0 h, 17 d vs 0 h, and 17 d vs 8 h.

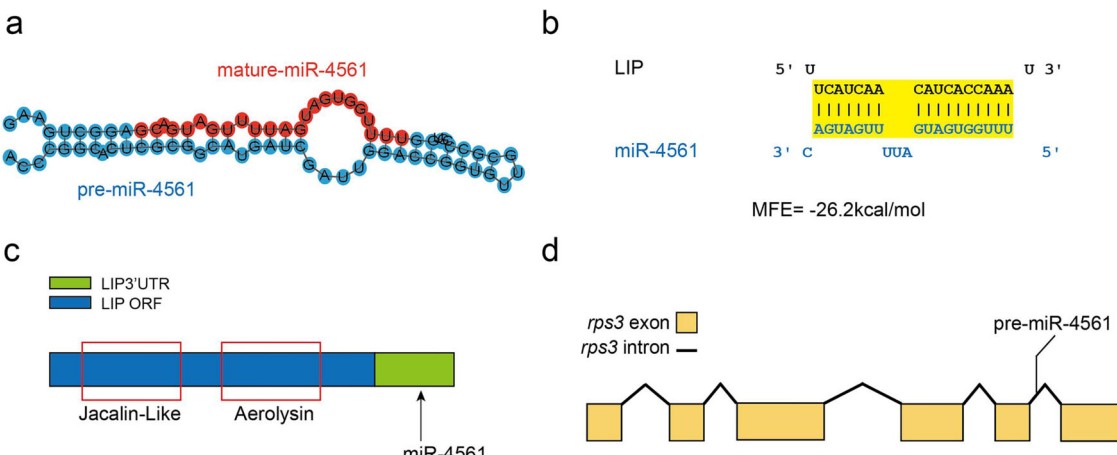

**Fig. 2 Characterization and target sequence prediction of miR-4561. a** Secondary structure of pre-miR-4561 and mature miR-4561 are marked in red. **b** miR-4561 and LIP mRNA combined sequence prediction. **c** LIP domain scheme. ORF is indicated by a blue box and 3′-UTR is shown in green. Jacalin and aerolysin domains are represented by red boxes. **d** Genomic location of pre-miR-4561.

that the expression of LIP increased over time following the infection (Fig. 3d, e). This is consistent with the role of LIP in the response to Gram-negative bacteria.

**miR-4561 inhibits LIP expression through binding to the 3′-UTR of LIP mRNA.** To validate the ability of miR-4561 to inhibit LIP expression in vitro, we performed a dual-luciferase reporter gene assay with plasmids encoding either a putative miR-4561 binding site or a mutant site (a 6-base-pair mutation in the seed region), transfected in cells overexpressing miR-4561 (Fig. 4a). The results showed that miR-4561 repressed luciferase activity when the reporter contained the wild-type *lip* 3′-UTR, but repression was lost when using the mutant *lip* 3′-UTR sequence (Fig. 4b).

We then tested whether miR-4561 interacted with *lip* mRNA in vivo, by generating an expression plasmid containing the EGFP

mRNA fused with either the entire WT LIP 3′-UTR or the mutant 3′-UTR. We then injected the plasmid in zebrafish embryos and detected EGFP expression after 1 day. We found that EGFP levels in embryos co-injected with WT EGFP-*lip* 3′-UTR and miR-4561 were significantly lower than when the EGFP reporter was injected alone (Fig. 4c). Moreover, we observed only a modest EGFP silencing in embryos co-injected with the mutated EGFP-*lip* 3′-UTR and miR-4561. We then further confirmed the EGFP fluorescence pattern by western blotting (Fig. 4d). These results indicate that *lip* is a target for miR-4561 and that miR-4561 targets the *lip* gene by binding to its 3′-UTR.

**Overexpression of miR-4561 inhibits *lip* in primary supra-neural body cells.** To investigate the function of miR-4561 in lamprey primary cells, we first transfected CY3-labeled miRNAs

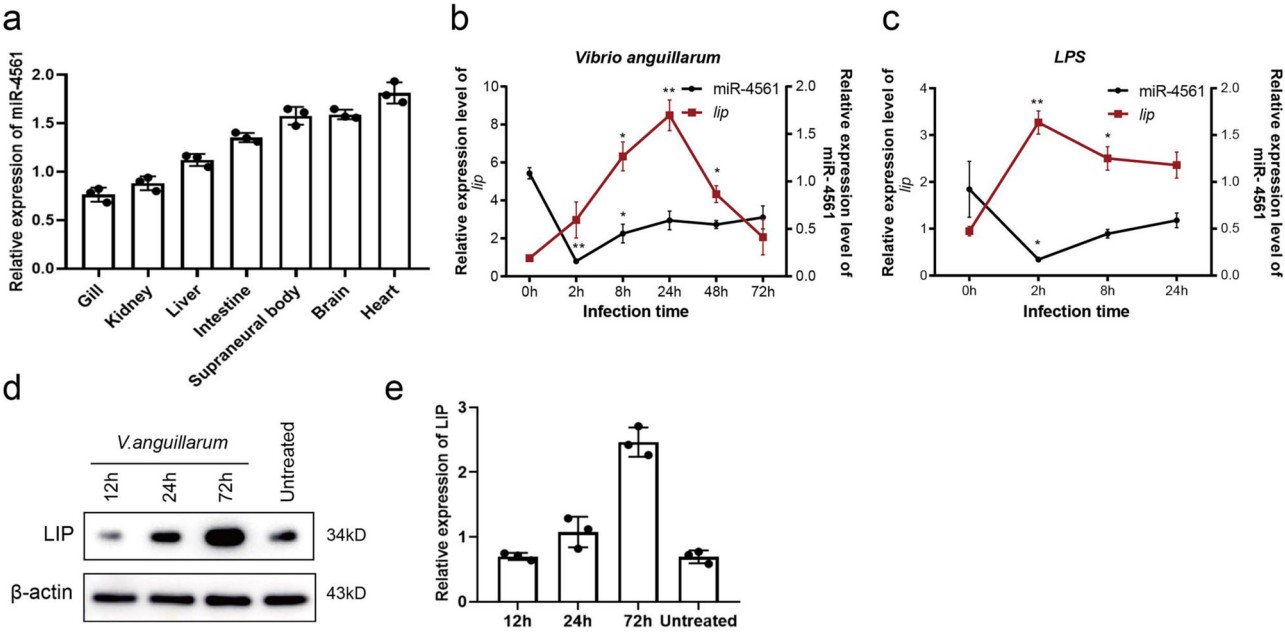

**Fig. 3 miR-4561 expression is down-regulated following bacterial infection. a** Relative expression of miR-4561 in lamprey tissues, as detected by qRT-PCR. **b, c** Time-course expression patterns of LIP and miR-4561 in lampreys infected with either *V. anguillarum* or LPS. **d** Time-course expression pattern of LIP protein in supraneural body tissue following bacterial infection. **e** LIP protein expression level in supraneural body tissues (*$P < 0.05$, **$P < 0.01$).

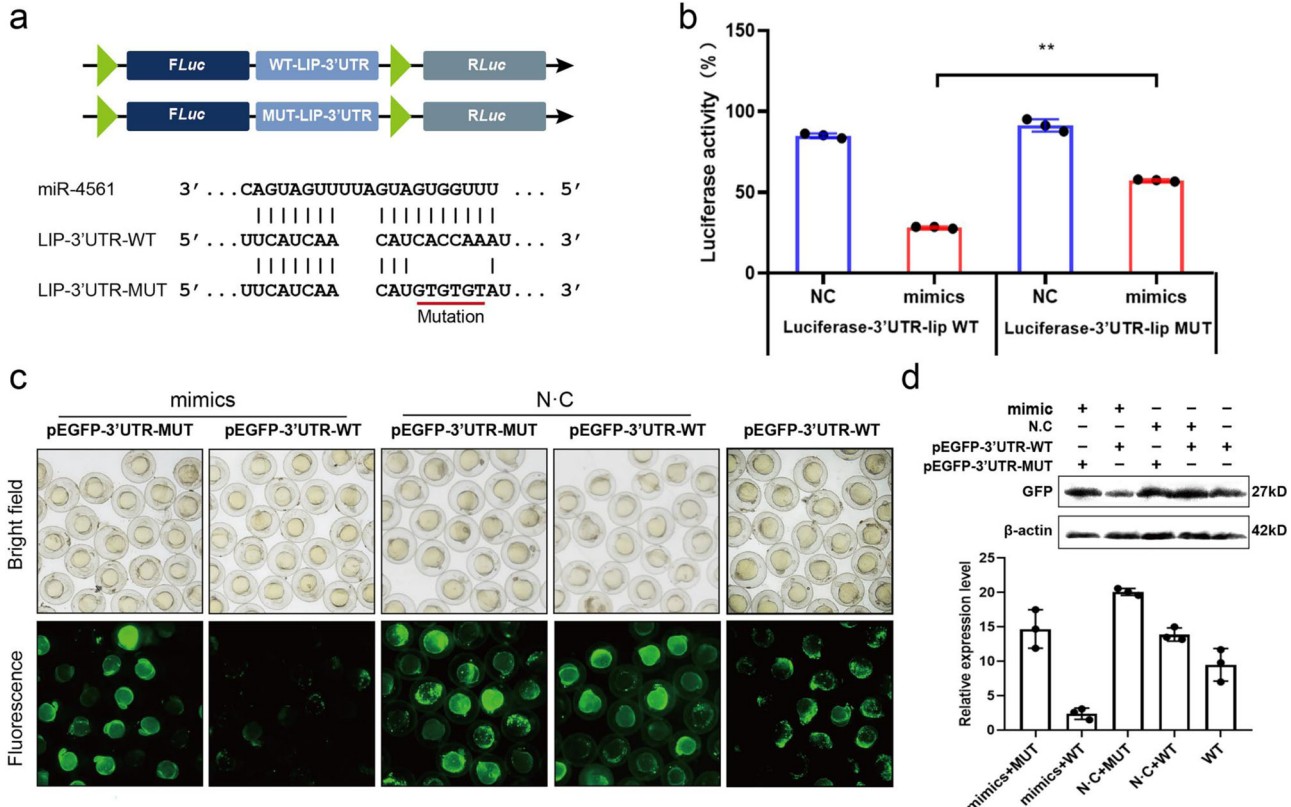

**Fig. 4 miR-4561 inhibits LIP expression through direct binding to the 3′-UTR of LIP mRNA. a** Nucleotide sequence of the miRNA target site on *lip* 3′-UTR. **b** Direct targeting of miR-4561 to *lip* 3′-UTR as detected by the dual-luciferase method. Wild-type and mutant miRNA target sequences were cloned into dual-luciferase reporters and co-transfected into HEK293 cells with a miR-4561 mimic. Each bar represents the relative luciferase activity (**$P < 0.01$). Values shown represent the mean ± SD of three independent experiments. **c** Zebrafish single-cell embryos were injected with an EGFP reporter fused with the 3′-UTR sequence, either wild-type or mutated in the presence or absence of miR-4561. Fluorescence levels were detected after 1 day. **d** Embryos treated as in (**c**) were lysed and the EGFP levels were measured by western blotting.

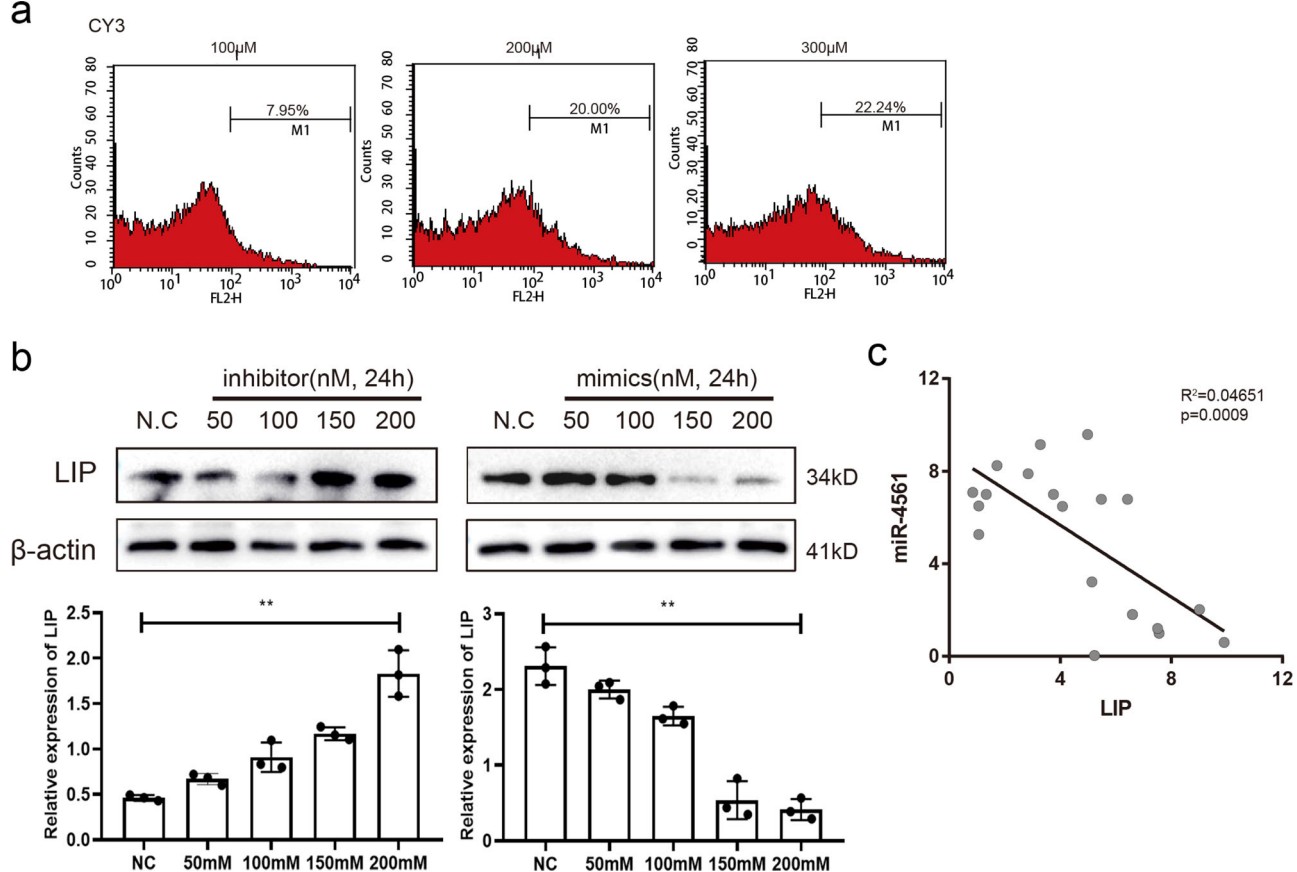

**Fig. 5 Overexpression of miR-4561 inhibits LIP expression in lamprey primary supraneural body cells. a** Detection of Cy3-labeled miRNA to gauge transfection efficiency. **b** miR-4561 mimic and inhibitor were transfected into primary supraneural body cells at the indicated concentration. After 24 h of transfection, LIP protein levels were measured by western blotting, using β-actin as loading control. Quantification of the blots is shown at the bottom ($^{**}P < 0.01$). **c** Linear regression analysis of the LIP mRNA and miR-4561 expression in 19 lamprey samples, as detected by qRT-PCR.

in supraneural body cells to find the optimal transfection concentration. We achieved the best transfection efficiency using 200 nM miRNA (Fig. 5a) and used this concentration in the subsequent experiments. We measured the effect of miR-4561 overexpression on LIP protein levels in supraneural body cells by western blotting. Results showed a significant decrease in LIP expression, compared to those observed in the controls ($P < 0.01$). Conversely, we did not observe the same trend when we co-transfected with a miR-4561 inhibitor (Fig. 5b). We also performed a linear regression analysis of the correlation between miR-4561 and LIP expression in 19 supraneural body tissue samples by performing qRT-PCR, using U6 as a normalizer. We observed that LIP mRNA levels were inversely correlated with miR-4561 levels (Fig. 5c, $P < 0.001$), consistent with the expression pattern observed in the sequencing analysis.

**miR-4561-mediated antibacterial regulation through *lip*.** We then tested the inhibitory effect of miR-4561 on LIP expression in vivo. We directly injected into the supraneural body tissues of live lampreys either a miR-4561 mimic, a miR-4561 inhibitor, or negative control (NC). We first measured inhibition efficiency. At 24 h after injection, we dissected the supraneural body tissue and analyzed, by quantitative RT-PCR, the expression level of miR-4561 and its target *lip*. Results showed that relative expression of miR-4561 significantly increased when we transfected the miR-4561 mimics, with a concomitant decrease of *lip* expression (Fig. 6a, b). Then we tested whether the overexpression of miR-4561 mimics affected the immune response to *V. anguillarum*.

We transfected either the miR-4561 mimics or the inhibitor, followed by *V. anguillarum* infection. We evaluated the antimicrobial effect of miR-4561 by determining the concentration of colony-forming units (CFU/mL) in the peritoneal lavage fluid harvested 48 h after the infection. Results showed that bacterial CFU of the peritoneal fluid harvested from lampreys transfected with the mimics was significantly higher than those from lamprey transfected with the inhibitor or the NC group, while the NC group had a higher number of CFU than the inhibitor group (Fig. 6c). Taken together, our results suggest that the overexpression of miR-4561 inhibits LIP expression and slows down bacterial clearance.

**miR-4561 regulates LIP expression and survival rate in lamprey embryos.** Subsequently, we determined the role of miR-4561 in lamprey embryonic development. First, we measured LIP mRNA expression level in every embryonic stage of normal embryos, observing that the peak of LIP mRNA expression was at the 256-cell stage (Fig. 7a). Then, we overexpressed miR-4561 mimics and inhibitor sequences in embryos and detected LIP expression level by immunohistochemistry (IHC). Results showed that the overexpression of mimics led to significantly lower LIP expression level compared to those observed when overexpressing the inhibitor, and in the control (uninjected embryos) (Fig. 7b).

Morphological observation of the microinjected embryos revealed that the embryos injected with the mimics were generally arrested at the 128-cell stage, although we did not observe any obvious difference in the embryos' external morphology (Fig. 7c).

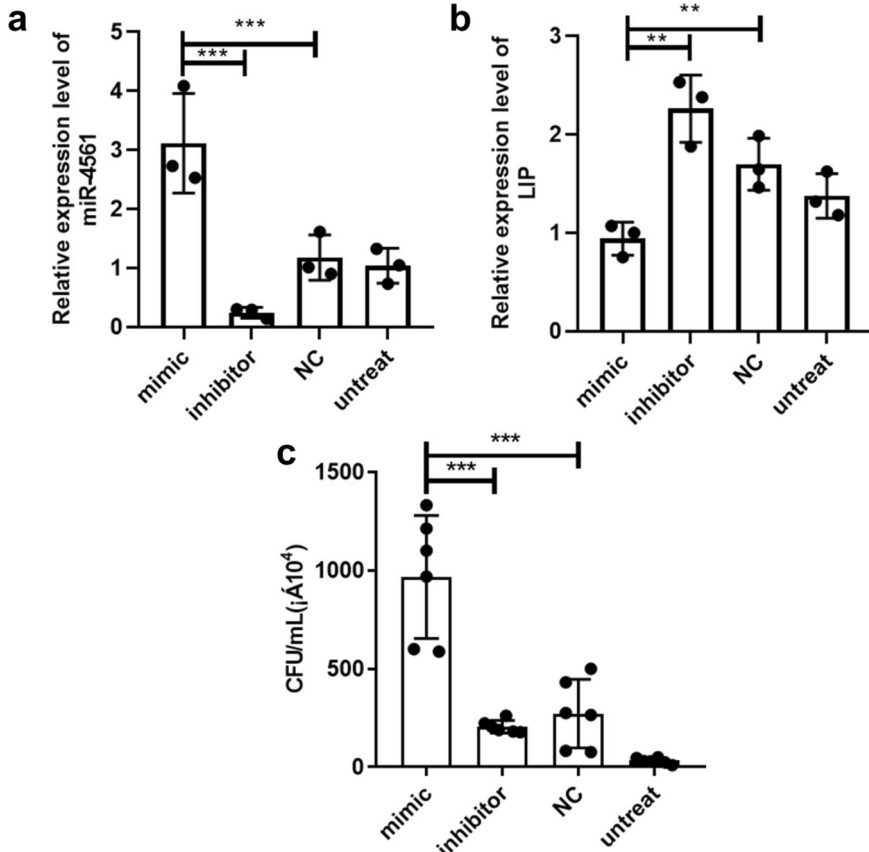

**Fig. 6 In vivo overexpression of miR-4561 mimics decreases *V. anguillarum* clearance. a** Relative expression level of miR-4561 after transfection in vivo. **b** Relative expression level of *lip*. **c** In vivo bacterial clearance after miR-4561 mimics and inhibitor transfection (**\*\***P < 0.01, **\*\*\***P < 0.001).

For this reason, we then used the point sampling method to count each batch of embryos and we measured the morphology and the survival status of embryos by microscopy. We microinjected 13 batches, 1000 embryos in each batch. Results showed that the hatching rate of embryos injected with miR-4561 mimics was significantly lower than that of the embryos injected with inhibitors (P = 0.042, n = 13 independent experiments). Moreover, the hatching rate of untreated embryos was higher than those reported for the other two groups (P = 0.068, n = 13 independent experiments) (Fig. 7d and Table S6).

We also analyzed pre-death embryos at the 128-cell stage using the terminal deoxynucleotidyl transferase dUTP nick end labeling (TUNEL) staining method, as a measure of apoptosis. Results showed that, although we observed no significant difference in the external shape of the 128-cell stage embryos, embryos injected with the miR-4561 mimics showed a significantly higher diffuse staining signal around the yolk granules compared to the signal observed in the empty injection group (Fig. 7e). These results suggest that the overexpression of miR-4561 caused developmental arrest and apoptosis.

**miRNA-4561 targeted LIP induces apoptosis through activation of TNF and NF-κB signaling pathways**. In order to further confirm that miRNA4561 targeting *lip* causes apoptosis in lamprey embryo development, three siRNAs (siRNA-LIP308, siRNA-LIP515, and siRNA-LIP942) were designed to silence lamprey *lip* gene expression. Figure 8a shows the fertilization rate after microinjection of lamprey and prove that the data can be used for subsequent experiments. Figure 8b represents a survival rate showing the neurula stage and head stage after microinjection of siRNA, respectively. In addition, the silencing effect of siRNA308

was higher than that of siRNA515 and siRNA942, and silencing *lip* gene expression significantly reduced after neurula stage. The embryo development map of lamprey after siRNA interference is given in Fig. 8c. TNF activates NF-κB mainly by TNFR pathways, whereas TNF binding to TNFR can induce apoptosis[18]. We consequently predicted that the *lip* gene loss-of-expression may play a role in the signal transmission pathway of persistent apoptosis in individuals. To verify the 10 genes in LIP–TNF–NF-κB signal axis through qRT-PCR, 10 pairs of primers were designed and the same RNA aliquots were assayed in triplicate (Fig. 8d). The results showed that *lip* gene silencing could induce apoptosis through activation of TNF and NF-κB signaling pathways. At the same time, we found that miR-4561 induces the activation of TNF and NF-κB signaling pathways and trigger apoptosis by targeting LIP (Fig. 8e). miR-4561 targeted *lip* gene loss-of-expression in lamprey results in apoptosis, cell growth arrest, and immune response, all of which serve to growth, development, and immune processes in lamprey.

**Discussion**

Leukocytes are normally involved both in the innate and adaptive immune responses. Therefore, it is of great importance to explore the role of miRNA in the regulation of leukocyte response to bacterial infection. In this study, we first generated three small RNA libraries from the leukocytes of lampreys infected with *V. anguillarum* and harvested at different times (0 h, 8 h, and 17 d) to identify differentially expressed miRNA. We identified a total of 21 conserved miRNAs and 61 unique novel miRNAs. Previous studies in other lamprey species using libraries generated from ammocoete larvae of the brook lamprey, adult Atlantic hagfish individuals, and nine sea lamprey organs (brain, gills, gut, heart,

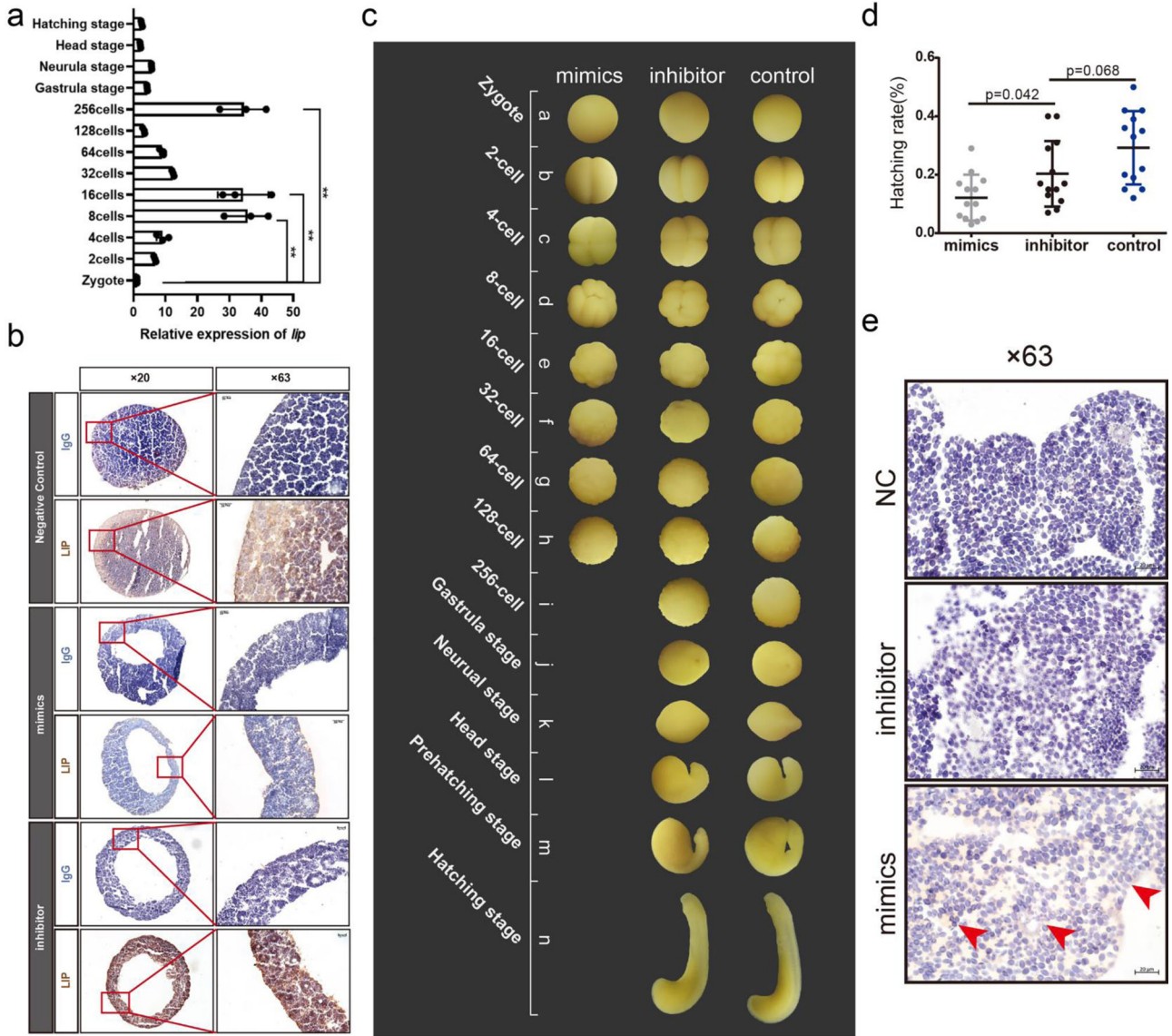

**Fig. 7 miR-4561 regulates LIP expression and survival rate in lamprey embryos. a** LIP mRNA expression level in normal lampreys at the indicated embryonic stage (**$P < 0.01$). **b** LIP protein staining using IHC of 256-cell stage embryos after microinjection of miR-4561 mimics, inhibitor, and NC. **c** Growth and development pattern of embryos after microinjection. **d** Survival rate of lamprey larvae recorded 11 d post-fertilization and hatching rate of lamprey embryos injected with miR-4561 mimics, inhibitor, and NC. **e** TUNEL staining of 128-cell stage embryos injected with miR-4561 mimics, inhibitor, and NC. The red arrow indicates the apoptotic signal.

kidney, liver, mouth, muscle, and skin) revealed that each organ expresses a specific miRNA pattern. The most expressed miRNA in the lamprey brain, gut, skin, kidney, liver, and muscle are miR-9a, miR-194, miR-205a, miR-30a-5, miR-122, and miR-1c, respectively. The four highest expressed miRNA genes in the heart are miR-30a-5, miR-30a-1, miR-26a, and miR-199a[15]. As assessed in this study, the six highest expressed miRNA genes in lamprey leukocytes are miR-4561, novel 2, miR-23, miR-29, miR-451, and novel 41, suggesting that both conserved miRNA families and lamprey-specific miRNAs could have very high expression level. Moreover, we measured the top 10 expressed miRNAs during bacterial infection (Fig. S1 and Table S7). The high abundance of these miRNAs suggests that they may play an important role in physiological processes such as the immune response. In lampreys, the supraneural body is the organ responsible for the hematopoietic activity, similarly to the bone marrow in higher vertebrates. In the supraneural body, all blood cells in all stages of maturity are present, including leukocytes and

their precursors[19]. In this study, we discovered a unique miRNA, miR-4561, which regulates the expression of LIP in leukocytes and the supraneural body.

LIP is an aerolysin-like protein similar to the lectin complex found in vertebrates[20,21]. LIP homologs can be found in frogs (βγ-CAT) and zebrafish (Dln1)[22,23]. LIP contains two functional domains, aerolysin and jacalin. Previous studies showed that aerolysin produced by *Aeromonas hydrophila* is necessary to trigger the NLRP3 inflammasome and necrotic cell death[24,25]. Conversely, previous studies reported that pineapple jacalin homologs can bind to CD45 to activate human T lymphocytes, leading to the activation of the ERK1/2 and p38 MAPK cascade and B lymphocyte apoptosis[26]. Recently, it has been reported that the sea anemone pore-forming toxin sticholysin II with anemone_cytotox domain inserts the N-terminus into the lipid core of cell membrane and assembles into a tetramer pore, resulting in the eukaryotic cell lysis[27]. Interestingly, LIP with Jaclin-like and ETX_MTX2 domains exhibits strong cytotoxic activity against

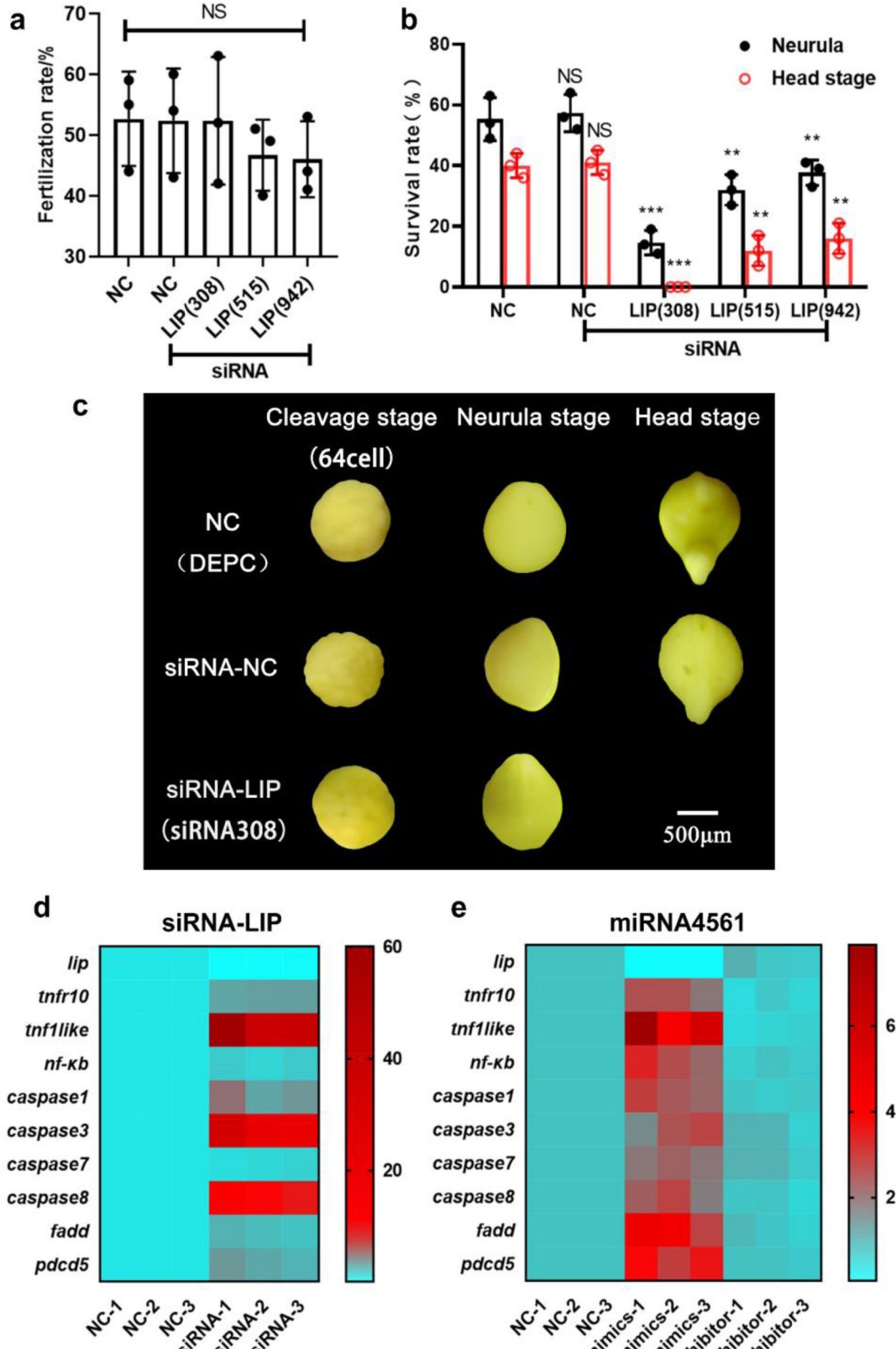

**Fig. 8 miR-4561 targeted LIP induces apoptosis through activation of TNF and NF-κB signaling pathways. a** Fertilization rate of siRNA silenced lamprey *lip* gene expression. **b** Survival rate of siRNA silenced lamprey *lip* gene expression in neurula stage and head stage (**$P < 0.01$, ***$P < 0.001$).
**c** Developmental map of siRNA silenced lamprey *lip* gene expression. After silencing the *lip* gene, lampreys are all lethal after neurula stage. **d** Changes of apoptosis-related genes in lamprey neurula stage after silencing *lip* gene by siRNA. **e** Changes of apoptosis-related genes in lamprey neurula stage after miRNA4561 targeting *lip* gene.

human tumor cells depending on the glycotype of the Glycosyl-phosphatidylinositol (GPI) anchors and the sphingomyelin on cell membrane[17], implying that the role of LIP in the lamprey's immune system is more complicated than previously thought.

In recent years, prediction methods to determine miRNA target genes have been improved by the use of algorithms, thereby greatly reducing the experimental burden. Software like miRanda and RNAhybrid use a different approach to identify candidate miRNAs: miRNA–mRNA binding thermal stability, sequence complementarity, sequence conservation, and RNA secondary structure[28]. By applying the same approach, we identified *lip* mRNA as a potential miR-4561 target. However, in silico predictions should be confirmed experimentally. Thus, we validated our discovery by using the dual-luciferase reporting system, a widely used technique to validate miRNA targets[29]. Then, we also validated the interaction in vivo by injecting an EGFP reporter system in zebrafish embryos[30]. Thus, the validation of our results, using different methods, reliably demonstrates that miR-4561 targets LIP mRNA and down-regulates LIP expression.

Several studies reported the existence of miRNAs, similar to miR-4561, which negatively regulate target gene expression following a bacterial infection. For example, miR-122 dynamically regulates TLR14 expression level of miiuy croaker (*Miichthys miiuy*) infected with *V. anguillarum*[31]. Moreover, miR-7a has been shown to regulate the expression of IRS2 and its downstream genes in flounders infected with *Edwardsiella tarda*[32]. The existence of this type of miRNA demonstrates that the continuous adjustment of the miRNA–mRNA interaction is crucial for the regulation of immune defense processes. Indeed, several miRNAs have been reported to bind to the 3′-UTR regions of different mRNAs encoding factors involved in the TLR signaling pathway. The binding negatively regulates target mRNA expression, either by inhibiting the translation process or by inducing mRNAs degradation[33,34]. However, in lampreys, both the TLR pathway and the downstream LIP signaling pathway are poorly understood. Thus, further studies are needed to uncover the mechanisms of the immune response in lampreys.

In this study, we also discovered that miR-4561 regulation of LIP affects embryonic development. We observed a significant decrease in the survival rate of embryos microinjected with miR-4561 mimics, compared to the uninjected embryos. Although this decrease could be caused by apoptosis induced by miR-4561 regulation of the immune pathway, we cannot exclude that this effect could be caused, at least in part, by mechanical damage or cytotoxicity caused by the microinjection process. It is important to point out that miRNA can target multiple genes at the same time. In this study, we identified eight potential miR-4561 target genes: myosin heavy chain 7 (MYH7), C-type lectin domain family 16 member A (CL16A), staphylococcal nuclease domain-containing protein 1 (SND1), ubiquitin-like modifier-activating enzyme 1 (UBA1), elongation factor 2 (EF2), tubulin alpha-1 (TBA1), Sec1 family domain-containing protein 1 (SCFD1), and LIP. Although these genes play an important role in development, immunity, apoptosis, and other vertebrate processes, whether or not these genes are targeted by miR-4561 in vivo needs further experimental validation[35–37].

In summary, miR-4561 activates the lamprey innate immune response by regulating LIP expression, thus mediating pathogen defense. The 3′-UTR of LIP mRNA is complementary to the miR-4561 seed region, and miR-4561 expression is negatively correlated to LIP expression. Moreover, miR-4561 may affect embryonic development. A schematic illustration of miR-4561 functions is shown in Fig. 9. This study provides useful information for understanding the LIP-mediated mechanisms of innate immune regulation in lampreys.

## Methods

**Animal feeding and *V. anguillarum* challenge**. Adult healthy lampreys (*L. morii*) (weight, 35 ± 5 g) were collected from the Yalu River and temporarily raised in a culture system composed of an aquarium (100 × 50 × 50 cm), black ceramsite sand filtration system (2–3 mm³), and ceramsite sand bottom layer (12 ± 2 cm high). Water height was about 40 cm and the water flow was about 2 L/min.

The *V. anguillarum* (28 °C) strain was cultured in 2216E liquid medium with 0.5% peptone, 0.1% yeast extract (pH = 8.0). A total of 0.1 mL of *V. anguillarum* (1 × 10⁷ cells/mL) and LPS (0.1 mg/mL) (Sigma Aldrich, St Louis, Missouri, USA) were individually injected into the abdomen of healthy lampreys (three for each group)[38–40]. Leukocytes were then separated by centrifugation using the Ficoll lymphocyte separation medium at a density of 1.092 (TBD, Tianjin, CHN). Supraneural body tissues were dissected and shredded into small chunks (1 cm³) with a 0.04% collagenase I (Sigma-Aldrich, USA) treatment.

All animal experiments were approved by the Animal Welfare and Research Ethics Committee of the Institute of Dalian Medical University and operated following the Guide for the Care and Use of Laboratory Animals 8th Edition Written by the National Research Council[41] (Permit Number: SCXK2008-0002).

**Total RNA extraction, small RNA library construction, and bioinformatics analysis**. Lamprey leukocyte total RNA was extracted using the RNAiso plus kit (TaKaRa, Dalian, CHN). After RNA quantification and identification, 3 µg of each RNA sample was used as the input material to prepare miRNA libraries. Sequencing libraries were generated using the NEBNext® Multiplex Small RNA Library Prep Set for Illumina (NEB, Ipswich, MA, USA) following the manufacturer's recommendations. PCR amplification was performed using the Long-Amp Taq 2× Master Mix kit, SR Primer for Illumina, and index (X) primer. PCR products were purified on an 8% polyacrylamide gel (100 V, 80 min). DNA fragments corresponding to 140~160 bp were recovered and dissolved in 8 µL of elution buffer. Finally, library quality was assessed on the Agilent Bioanalyzer 2100 system using DNA High Sensitivity Chips (Agilent, Santa Clara, CA, USA).

**LIP mRNA-interacting miRNA prediction**. The ability of differentially expressed miRNA to target LIP mRNA was evaluated using the miRanda and RNAhybrid algorithms using sequence complementation, RNA secondary structure, and thermal stability of miRNA–mRNA.

**Stem-loop RT-PCR and quantitative RT-PCR**. Adult lampreys were infected with *V. anguillarum* and cultured in separate tanks. Total RNA from different tissue and cell samples were extracted using the RNAiso Plus kit (TaKaRa, CHN), and then, samples were treated with DNase I to remove the residual genomic DNA. Total RNA samples were then reverse transcribed to cDNA using the miRNA First Strand cDNA Synthesis SuperMix (TransGen, Beijing, CHN) with added stem-loop (RT) primers (Table S1), following the manufacturer's instructions. The relative miRNAs expression level were determined using the miRNA Green qPCR SuperMix (TransGen, Beijing, CHN) on the ABI7500 Real-Time PCR System (Applied Biosystems, USA). We used a 20 µL qPCR mixture consisting of 2 µL of cDNA, 10 µL of 2× Green qPCR SuperMix, 0.4 µL of 50× Passive Reference Dye, 6.8 µL of nuclease-free H₂O, 0.5 µL of universal miRNA qPCR primer (10 µM), and 0.4 µL of forward primer. The PCR reaction was as follows: 94 °C for 30 s, followed by 40 cycles at 94 °C for 5 s, 60 °C for 15 s, and 72 °C for 10 s. Data were normalized using the U6 reference gene. At least the biologically independent experiments were performed.

Detection of gene expression, other than miRNA, was performed using quantitative reverse transcription PCR (qRT-PCR). The total qRT-PCR reaction volume was 20 µL and composed of 2 µL of template cDNA, 0.4 µL of forward primer (10 µM) and reverse primer (10 µM) (Sangon, Shanghai, CHN), 10 µL of 2× TOP Green qPCR Supermix, 7.2 µL of nuclease-free H₂O. The PCR reaction was as follows: 94 °C for 30 s, followed by 40 cycles at 94 °C for 5 s, 60 °C for 15 s, and 72 °C for 10 s. GAPDH was used as reference gene and the relative expression was calculated by the $2^{-\triangle\triangle CT}$ method. Primer sequences used for qRT-PCR are shown in Table S1.

**Western blotting**. Total proteins were extracted from lamprey tissues and protein concentrations were determined using a BCA Protein Assay kit. Protein samples were loaded into a 15% polyacrylamide gel and transferred onto a nitrocellulose membrane. Membranes were incubated for 2 h in 5% nonfat dried milk to block non-specific epitopes. Then, membranes were incubated with the primary antibody (0.5 µg/mL) overnight at 4 °C. Membranes were then washed four times (10 min each) in Tris-buffered saline-Tween-20 (TBST). Membranes were incubated with HRP-conjugated anti-rabbit IgG and anti-mouse IgG antibodies (Abcam, Cambridge, MA, USA) used at 1:5000 dilution (50 ng/mL). Protein bands were visualized using enhanced chemiluminescence (ECL) reagent (Proteintech, Rosemont, IL, USA). β-actin was used as loading control for western blotting. Anti-β-actin antibody was purchased from Abcam (USA) while anti-LIP antibody was produced in our laboratory.

**Reporter plasmids generation for luciferase assays**. The 3′-UTR LIP mRNA wild-type and mutated sequences (GenBank accession No. MT176433) were cloned

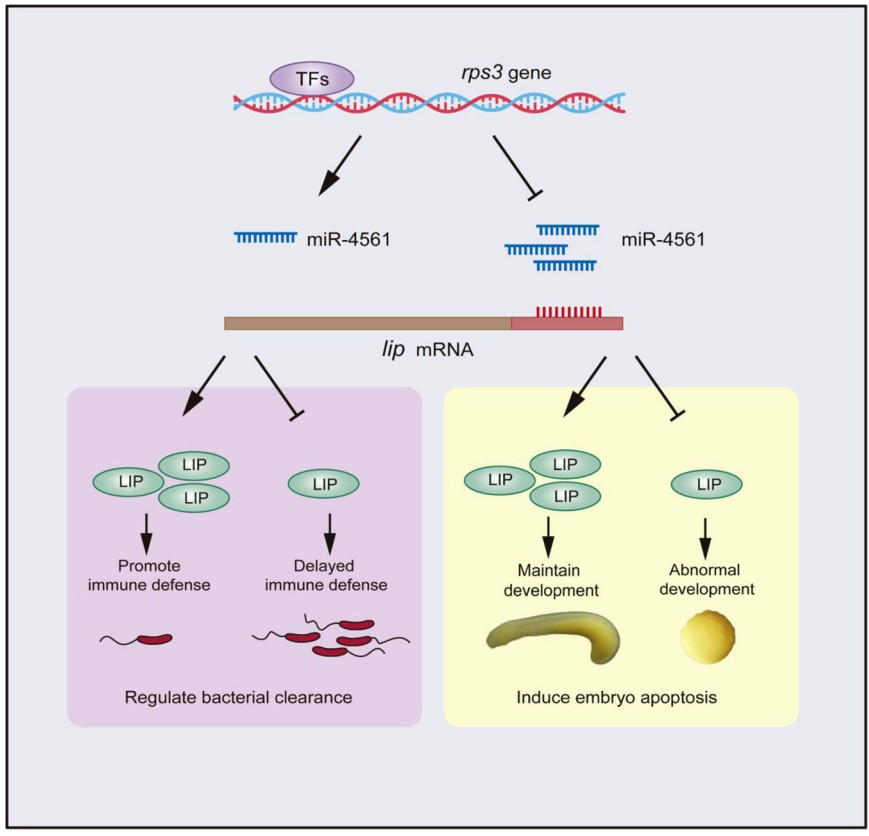

**Fig. 9 Regulatory mechanism of miR-4561.** Schematic representation of miR-4561-mediated regulation of bacterial clearance and embryo apoptosis through the suppression of the *lip* gene.

into the pmirGLO control vector (Promega, Madison, WI, USA) to obtain the pmirGLO-*lip*-3′-UTR-WT/MUT plasmids, respectively. The same sequences were cloned into the pEGFP-C1 vector to obtain the pEGFP-*lip*-3′-UTR-WT/MUT plasmids, respectively.

HEK293 cells were seeded on 48-well plates overnight before transfection and then transfected with 25 ng of dual-luciferase reporter WT/MUT vector and 50 nM of miRNA mimics or inhibitor (GenePharma, Suzhou, CHN). Dual-luciferase reporter assays were performed 24 h after transfection using the luciferase activity assay kit, following the manufacturer's instructions (Beyotime, Beijing, CHN). Firefly luciferase activity was normalized to Renilla luciferase activity. Relative light unit (RLU) data were measured using a microplate reader (Molecular Devices, San Jose, CA, USA) in luminescence detection mode.

**Cell culture and transient transfection.** Supraneural body tissues were dissected from lamprey and digested with 0.04% collagenase I (Sigma Aldrich, USA) for 10 min at 18 °C to collect $1 \times 10^6$ cells/mL primary cells. Primary supraneural body cells were cultured in L-15 medium (Sigma Aldrich, USA) at 18 °C. HEK293 cells were cultured in RPMI-1640 medium (Hyclone, South Logan, UT, USA) supplemented with 10% fetal bovine serum (Hyclone) in a humidified atmosphere containing 5% $CO_2$ at 37 °C. HEK293 cells were inoculated into 24- or 96-well plates and cultured overnight. Plasmids and miRNAs were transiently transfected with Lipofectamine 3000 (Invitrogen, Carlsbad CA, USA), following the manufacturer's instruction. miR-4561 mimics (5′-UUUGGUGAUGAUUUUGAUGACG-3′), miR-4561 inhibitor (5′-CGTCATCAAAATCATCACCAAA-3′), and the NC (5′-UUC UCCGAACGUGUCACGUUT-3′) were synthesized by GenePharma (Suzhou, CHN). Supraneural body cells and HEK293 cells were transfected with 30–200 nM of each oligonucleotide.

**MiRNA mimics and inhibitor microinjection assays.** Zebrafish embryos were injected into the blastodisc at the 1- to 2-cell stage using a micromanipulator (Narishige, Setagaya, Japan). The injection doses were 50 μM 4–5 nL/embryo. For co-injection experiments, samples were mixed thoroughly before injection. Each embryo was injected with 1.2 ng of mimics and 400 pg of pEGFP-lip-3′-UTR-WT/MUT plasmids

**In vivo overexpression of miRNAs.** miR-4561 mimics, inhibitor, and NC were transfected into supraneural body tissues using the Entranster (Engreen, Beijing, CHN) reagent, containing 10% glucose, according to the manufacturer's instructions.

The miR-4561 mimics (5′-UUUGGUGAUGAUUUUGAUGACG-3′), miR-4561 inhibitor (5′-CGTCATCAAAATCATCACCAAA-3′), and the NC (5′-UUC UCCGAACGUGUCACGUUT-3′) were synthesized by GenePharma (CHN). Oligonucleotides were diluted in DEPC water at a final concentration of 50 μM and then mixed with the transfect reagent. Transfection efficiency was evaluated by qRT-PCR of miR-4561 and lip.

**_V. anguillarum_ CFU determination.** According to our previous study, after 12 h of miRNA transfection, 100 μL of $1 \times 10^7$ cells/mL of *V. anguillarum* was intraperitoneally injected into lampreys[42]. After 24 h, 1 mL of PBS was used to irrigate the abdominal cavity and an equal volume of peritoneal lavage fluid was collected, diluted $2 \times 10^4$ times with PBS, and cultured in an LB plate overnight at 28 °C. Each experiment was repeated three times.

**Immunohistochemistry (IHC) assay.** Paraffin sections were preheated, dewaxed in xylene, and rehydrated with a gradient ethanol concentration. Antigen retrieval buffer was used to induce epitope recovery. Slices were then incubated in a 0.3% $H_2O_2$ solution for 15 min to block endogenous peroxidase. Then, slices were incubated with a 2% normal goat serum for 2 h to block non-specific antigenic determinants. The anti-LIP antibody (0.5 μg/mL) was then added overnight at 4 °C. Then, slices were incubated with the secondary antibody for 30 min in the dark and treated with DAB. After performing hematoxylin staining for 10 min, slices were dehydrated via gradient concentration in ethanol and xylene, then mounted with neutral balsam. LIP staining rate was calculated according to the percentage of positive cells per microscope field and normalized on the total number of cells in each field.

**Analysis of lamprey embryonic development and TUNEL assay.** Microinjection and culture methods for lamprey embryos were performed as previously described[6,38]. In brief, embryos were cultured in clean aquaculture water at 6–18 °C. Fertilized eggs were kept 1 cm apart and oxygenated. Embryos were then randomly collected for microinjection, nucleic acid analysis, and TUNEL experiments.

For TUNEL, frozen sections of embryo were washed twice in PBS and incubated with 2% hydrogen peroxide for 5 min at room temperature to inactivate endogenous peroxidases. TdT reaction and DAB color development were carried out according to the manufacturer's instructions (GeneCopoeia, Rockville, MD,

USA). Thereafter, slices were mounted with neutral balsam. The apoptosis signal was then observed using light microscopy.

**siRNA delivery for lamprey embryos**. Three groups of 100 fertilized embryos were injected with *lip* siRNA (Shanghai GenePharma Co., Ltd. RNAi duplex with sense sequence: 5′ CCGCAACCGUGAGUUCUUUTT3′ and antisense sequence: 5′ AAAGAACUCACGGUUGCGGTT3′) or NC (RNAi duplex with sense sequence: 5′ UUCUCCGAACGUGUCACGUTT3′, and antisense sequence: 5′ACGUGACAC GUUCGGAGAATT3′). One- or two-cell stage embryos were injected with 10 nL of each siRNA at 40 mM concentration (dissolved in DEPC water). Embryos were reared in fish water at 18 °C. Embryos were monitored to assess whether they developed normally, beyond the blastula stage. And the survival rates of neurula stage and head stage embryo were calculated. Embryos were snap-frozen in liquid nitrogen and stored at −80 °C before processing for qRT-PCR. RNA was extracted from 30 embryos on day 6 post-injection.

**Statistical analysis**. GraphPad Prism 6 was used for statistical analysis. Statistically significant differences were calculated using Student's *t*-test and one-way analysis of variance, as appropriate. The following *P*-values were considered significant and are indicated with asterisks in the figures: $^*P < 0.05$, $^{**}P < 0.01$, $^{***}P < 0.001$.

**Ethics approval and consent to participate**. The animal experiments were performed in accordance with the regulations of the Animal Welfare and Research Ethics Committee of the Institute of Dalian Medical University's Animal Care protocol (Permit Number: SCXK2008-0002).

**Reporting summary**. Further information on research design is available in the Nature Research Reporting Summary linked to this article.

## Data availability
All relevant data are available from the corresponding author upon reasonable request. Source data underlying plots shown in figures are provided in Supplementary information. All miRNA transcriptome data have been uploaded with the SRR accession numbers SRR11306709, SRR11306708, and SRR11306707. All source data underlying the graphs and charts shown in the main and supplementary figures are presented in Supplementary Data 1.

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

## Acknowledgements

This work was funded by Chinese National Natural Science Foundation Grants (No. 31772884, No.32070518). The project of Department of Ocean and Fisheries of Liaoning Province (No. 201805), Program of Science and Technology of Liaoning Province (No. 2019-MS-218), and Science and Technology Innovation Fund Research Project (No. 2018J12SN079).

## Author contributions

Y.P. and L.M. designed research; L.M., Z.D., and T.Z. performed research; J.L. contributed lamprey material/microinjection methods; Y.P., L.M., and Q.W.L. analyzed data and prepared figures; and Y.P., L.M., and M.G. wrote the paper.

## Competing interests

The authors declare no competing interests.
