## [Peer Review File. · Communications Biology]

Reviewers' comments:

Reviewer #1 (Remarks to the Author):

Lie et al., described the microRNA expression profile in lamprey, but there is less mention on the aspects of comparative immunological reasons, or the advantage/disadvantage why the authors selected these kind of jawless animals. If the authors analyzed the data in the view of comparative aspects that would help to understand the role and biology of miR-4561.

It is very difficult to understand the reasons why the authors used *V. anguillarum* (*L. anguillarum*) as an infection agent. As far as literature reviews, there has been no report that *L. anguillarum* is pathogenic to lamprey. So it needs to give full explanations in detail. Since if it is a pathogenic agent, the lamprey should exhibit clearly different immune response compared with just foreign substance. This fact leads that the authors need to have different view in terms of data analysis and discussion. Another thing is that there is no code No, in the animal welfare approval.

Reviewer #2 (Remarks to the Author):

The authors founded that miR-4561 could bind to the 3'UTR of Lamprey immune protein (LIP) and regulate the expression of LIP and antibacterial reaction. This is of great significance for the regulation of LIP function. Some methods in this manuscript are useful for identifying of other miRNA and target genes.

My specific comments are as follows:

Question 1:

Line 78-79: "the gram-negative bacterium *V. anguillarum*", should revised to "the Gram-negative bacterium *V. anguillarum*", full text unified.

Question 2:

Line 106: Figure 1. "Small RNA annotation" should revised to "Small RNA type composition".

Question 3:

Line 117: "We found 60 miRNAs, of which 13 were up-regulated or down-regulated between 0 h and 8 h; 24 were down-regulated between 0 h and 17 d, and 23 up- or down-regulated between 8 h and 17 d. The expression levels of these miRNAs displayed a high degree of variation, ranging from 1 to 1356331 (Tables S1-S3). We identified two miRNAs, miR-4561 and miR-novel-14, which were differentially expressed among all groups (Figure 1D)." should revised to "We found 60 miRNAs, of which 13 were up-regulated or down-regulated between 0 h and 8 h; 24 were down-regulated between 0 h and 17 d, and 23 up- or down-regulated between 8 h and 17 d (Figure 1D). The expression levels of these miRNAs displayed a high degree of variation, ranging from 1 to 1356331 (Tables S1-S3). We identified two miRNAs, miR-4561 and miR-novel-14, which were differentially expressed among all groups."

Question 4:

Line 140: "miR-4561 expression levels decreased markedly after 2 h, reaching the lowest levels after 8 and 24 h. At 72 h, the expression levels recovered as before the stimulation." According to Figure 3B, the miR-4561 expression level reached the lowest at 8h and did not appear to return to pre-experimental expression levels at 72 h. In addition, the expression level suggested not to use the plural form, the full text unified.

Question 5:

Line 148: Figure 3, The stripes of β -actin in figure 3D are not consistent. These questions also exit in Figure 5.

Line 151-153: "(D) Time-course expression pattern of TNFR in supraneural body tissue following bacterial infection. (E) LIP protein expression levels in supraneural body tissues. (F) Quantification

of the blot shown in panel (E).” , The (D) is irrelevant content. This needs to check carefully. And there are no figure 3E.

Question 6:

Line 159: “Wild type”, should revised to “wild type”.

Question 7:

Line 220: Figure 6 “Relative expression level of miR-4561 after transfection in vivo.” Suggest to add the expression level of miR-4561 in the inhibitor group.

Question 8:

Line 252: Figure 7 “TUNEL staining of 128-cell stage embryos injected with miR-4561 mimics, inhibitor, and NC.” Please add a description to the callout about the red arrow in the figure.

Question 9:

Line 330: Figure 8 The schematic “Inducing embryo apoptosis” should revised to “Induce embryo apoptosis”.

Question 10:

Line 335: Animal feeding and *V. anguillarum* challenge section, what’s the water temperature during the adult healthy lampreys temporarily raised? Why choose the *V. anguillarum* bacteria to challenge the lampreys? What are the symptoms of *V. anguillarum* infection?

Question 11:

Line 340: “A total of 0.1 mL of *V. anguillarum* (1×10^7 CFU/mL) and LPS (0.1 mg/mL) (Sigma Aldrich, St. Louis, Missouri, USA) were individually injected into the abdomen of healthy lampreys (three for each group).” Please add here the basis for the concentration of the bacterial infection or pre-experimental description.

Question 12:

Line 418: “Supraneural body tissues were dissected from lamprey and digested with 0.04% collagenase I (Sigma Aldrich, USA) for 10 min at 18 °C to collect 1×10^6 cells/mL primary cells.” Should revised to “Supraneural body tissues were dissected from lamprey and digested with 0.04% collagenase I (Sigma Aldrich, USA) for 10 min at 18 °C to collect 1×10^6 cells/mL primary cells.”

Question 13:

Line 428: “Supraneural body cells and HEK293 cells were transfected with 30–200 nM of each oligonucleotide.”, confirm the HEK293T or HEK293.

Question 14:

Line 472-473: “p-values were considered significant and are indicated with asterisks in the figures: * $P < 0.05$, ** $P < 0.01$, *** $P < 0.001$.”, confirm the “*** $P < 0.01$, ** $P < 0.001$ ”, they are same now.

Reviewer #3 (Remarks to the Author):

This article investigated whether microRNAs play a key regulatory role in the immunity of lamprey. The authors studied the changes of microRNAs expression in leukocytes of lamprey following *Vibrio anguillarum* infection. Using comparative methods, they identified some microRNAs potentially involved in immune regulation in lamprey. Among these microRNAs, they discovered that lamprey miR-4561 targets LIP mRNA and downregulates the LIP expression. miR-4561 not only participates in the lamprey innate immune response, but also affects embryonic development.

This paper showed lamprey microRNAs may significantly affect cellular immunity and apoptosis by regulating gene expression. This is the first study to investigate microRNAs implicated in the antibacterial defense of lamprey. This work is meaningful and valuable for further research on the

functions of microRNAs in jawless vertebrates. However, there are some shortcomings:

1. While the authors observed that overexpression of miR-4561 induced apoptosis in embryonic cells, inhibition of miR-4561 insignificantly affect embryonic development. Additionally, the authors did not examine whether the expression of miR-4561 changes during embryonic development. Thus, the authors need to conduct further experiments to gather more evidences to confirm that miR-4561 is indeed involved in embryonic development under physiological conditions.

2. There are some mistakes in this article. The descriptions of Figure 3 are inaccurate in line 151-153. The authors described "either a miR-4561 mimic or inhibitor" in line 175, but only a miR-4561 mimic was used in Figure 4B. "The expression of miR-4561 was normalized by U6 RNA" should be deleted in line 177.

In conclusion, this work is meaningful but the details of the work needs some improvements.

Reviewer #1 (Remarks to the Author):

Lie et al., described the microRNA expression profile in lamprey, but there is less mention on the aspects of comparative immunological reasons, or the advantage/disadvantage why the authors selected these kind of jawless animals. If the authors analyzed the data in the view of comparative aspects that would help to understand the role and biology of miR-4561.

It is very difficult to understand the reasons why the authors used *V. anguillarum* (*L. anguillarum*) as an infection agent. As far as literature reviews, there has been no report that *L. anguillarum* is pathogenic to lamprey. So it needs to give full explanations in detail. Since if it is a pathogenic agent, the lamprey should exhibit clearly different immune response compared with just foreign substance. This fact leads that the authors need to have different view in terms of data analysis and discussion. Another thing is that there is no code No, in the animal welfare approval.

Response :

Thank you for your question.

V. anguillarum belongs to the non-dominant flora in the intestinal tract of *Lampetra japonica*(please refer to the reference 1 below). Under normal conditions, intestinal symbiotic flora in balance does not cause host immune response. *L. anguillarum* isolated in vitro is an antigen for lampreys, and then could induce immune stimulation in lampreys. Besides, *V. anguillarum* has been detected in the seawater where we fished for lamprey. As you can see, in this article and in our previous papers, we have used a concentration of 1×10^7 CFU/mL, which can already produce an immune response. In previous experiments, we also tried the concentration of 1×10^8 CFU/mL with a stronger immune response, but it would cause a certain number of fish deaths. For the purpose of reducing animal deaths, we set the stimulation concentration to the 1×10^7 CFU/mL.

Although there is no evidence that *V. anguillarum* is the pathogen of lamprey, it can indeed trigger a series of immune responses as a gram-negative bacterium (please refer to the reference 2-6). For example, the expression of lamprey serum apolipoprotein in lamprey leukocytes and sera increased significantly at different times after stimulating *Vibrio anguillarum* (2); the expression levels of L-caveolin

exhibited consistent increases in expression in leukocytes stimulated by *Vibrio anguillarum* (3); the mRNA expression level of L-serpin increased significantly after *Vibrio anguillarum* and dramatically peak at 8 h(4); Lamprey Complement factor I and TNFR10 were also significantly up-regulated under the stimulation of *Vibrio anguillarum* (5,6). The reason we prefer to use this bacteria is that it can be used as a double verification of Gram-negative bacteria (LPS) stimulation.

1. Li Y, Xie W, Li Q. Characterisation of the bacterial community structures in the intestine of *Lampetra morii*. *Antonie Van Leeuwenhoek*. 2016 Jul;109(7):979-86. doi: 10.1007/s10482-016-0699-0. Epub 2016 May 14. PMID: 27180095.
2. Han Q, Han Y, Wen H, Pang Y, Li Q. Molecular Evolution of Apolipoprotein Multigene Family and the Original Functional Properties of Serum Apolipoprotein (LAL2) in *Lampetra japonica*. *Front Immunol*. 2020 Aug 11;11:1751. doi: 10.3389/fimmu.2020.01751. PMID: 32849624; PMCID: PMC7431520.
3. Teng H, Wang D, Lu J, Zhou Y, Pang Y, Li Q. Novel insights into the evolution of the caveolin superfamily and mechanisms of antiapoptotic effects and cell proliferation in lamprey. *Dev Comp Immunol*. 2019 Jun;95:118-128. doi: 10.1016/j.dci.2019.01.005. Epub 2019 Feb 8. PMID: 30742851.
4. Wang D, Gou M, Hou J, Pang Y, Li Q. The role of serpin protein on the natural immune defense against pathogen infection in *Lampetra japonica*. *Fish Shellfish Immunol*. 2019 Sep;92:196-208. doi: 10.1016/j.fsi.2019.05.062. Epub 2019 Jun 5. PMID: 31176010.
5. Lv W, Ma A, Chi X, Li Q, Pang Y, Su P. A novel complement factor I involving in the complement system immune response from *Lampetra morii*. *Fish Shellfish Immunol*. 2020 Mar;98:988-994. doi: 10.1016/j.fsi.2019.11.017. Epub 2019 Nov 8. PMID: 31712129.
6. Zhu Y, Li J, Li Q, Pang Y. Characterization of lamprey (*Lampetra japonica*) tnfr10-like gene: A potential granulocyte marker molecule and its immune functions. *Mol Immunol*. 2020 Aug;124:25-34. doi: 10.1016/j.molimm.2020.05.015. Epub 2020 Jun 1. PMID: 32497752.

Besides, we have added the animal welfare approval number that was missed before. Please refer to Page 17, Line 375-376 and Page 22, 542-544.

Reviewer #2 (Remarks to the Author):

The authors founded that miR-4561 could bind to the 3'UTR of Lamprey immune protein (LIP) and regulate the expression of LIP and antibacterial reaction. This is of great significance for the regulation of LIP function. Some methods in this manuscript are useful for identifying of other miRNA and target genes.

My specific comments are as follows:

Question 1:

Line 78-79: “the gram-negative bacterium *V. anguillarum*”, should revised to “the Gram-negative bacterium *V. anguillarum*” , full text unified.

Response 1:

Thank you for the comment. We have revised “the gram-negative bacterium *V. anguillarum*” to “the Gram-negative bacterium *V. anguillarum*”. Please refer to Page 3, Line 78-79.

Question 2:

Line 106: Figure 1. “Small RNA annotation” should revised to “Small RNA type composition”.

Response 2:

Thank you for the comment. We have revised “Small RNA annotation” to “Small RNA type composition”. Please refer to Page 4, Line 106.

Question 3:

Line 117: “We found 60 miRNAs, of which 13 were up-regulated or down-regulated between 0 h and 8 h; 24 were down-regulated between 0 h and 17 d, and 23 up- or down-regulated between 8 h and 17 d. The expression levels of these miRNAs displayed a high degree of variation, ranging from 1 to 1356331 (Tables S1-S3). We identified two miRNAs, miR-4561 and miR-novel-14, which were differentially expressed among all groups (Figure 1D).” should revised to “We found 60 miRNAs, of which 13 were up-regulated or down-regulated between 0 h and 8 h; 24 were down-regulated between 0 h and 17 d, and 23 up- or down-regulated between 8 h and 17 d (Figure 1D). The expression levels of these miRNAs displayed a high degree of variation, ranging from 1 to 1356331 (Tables S1-S3). We identified two miRNAs, miR-4561 and miR-novel-14, which were differentially expressed among all groups.”

Response 3:

We have revised “We found 60 miRNAs, of which 13 were up-regulated or down-regulated between 0 h and 8 h; 24 were down-regulated between 0 h and 17 d, and 23 up- or down-regulated between 8 h and 17 d. The expression levels of these miRNAs displayed a high degree of variation, ranging from 1 to 1356331 (Tables S1-S3). We identified two miRNAs, miR-4561 and miR-novel-14, which were differentially expressed among all groups (Figure 1D).” to “We found 60 miRNAs, of which 13 were up-regulated or down-regulated between 0 h and 8 h; 24 were down-regulated between 0 h and 17 d, and 23 up- or down-regulated between 8 h and 17 d (Figure 1D). The expression level of these miRNAs displayed a high degree of variation, ranging from 1 to 1356331 (Tables S1-S3). We identified two miRNAs, miR-4561 and miR-novel-14, which were differentially expressed among all groups.” Please refer to Page 5, Line 117-122.

Question 4:

Line 140: “miR-4561 expression levels decreased markedly after 2 h, reaching the lowest levels after 8 and 24 h. At 72 h, the expression levels recovered as before the stimulation.” According to Figure 3B, the miR-4561 expression level reached the lowest at 8h and did not appear to return to pre-experimental expression levels at 72 h. In addition, the expression level suggested not to use the plural form, the full text unified.

Response 4:

We apologize for the inaccurate description here. We have revised it to “ the expression level of miR-4561 decreased significantly after 2 hours, and began to recover after 8 hours. At 72 hours, the expression level recovered significantly but was still lower than the pre-experimental expression levels. ” Please refer to Page 5, Line 140-142.

Besides, we have unified the expression level as singular throughout the full text.

Question 5:

Line 148: Figure 3, The stripes of β -actin in figure 3D are not consistent. These questions also exist in Figure 5.

Line 151-153: “(D) Time-course expression pattern of TNFR in supraneural body tissue following bacterial infection. (E) LIP protein expression levels in supraneural body tissues. (F) Quantification of the blot shown in panel (E).” , The (D) is

irrelevant content. This needs to check carefully. And there are no figure 3E.

Response 5:

Thank you for your question. We have completely repeated the related WB experiments in Figure 3D and Figure 5 and solved the problem. Please check Figure 3D and Figure 5.

We are really sorry for our carelessness. We used the wrong version of the picture before, which caused the legend and the figure to be inconsistent. We have fixed this problem now, please refer to Page 6, Line 148-152.

Question 6:

Line 159: “Wild type”, should revised to “wild type”.

Response 6:

Thank you for the comment. We have revised “Wild type” to “wild type”. Please refer to Page 6, Line 158.

Question 7:

Line 220: Figure 6 “Relative expression level of miR-4561 after transfection in vivo.” Suggest to add the expression level of miR-4561 in the inhibitor group.

Response 7:

Thank you for your question. We have added the expression level of miR-4561 in the inhibitor group, please refer to Figure 6A.

Question 8:

Line 252: Figure 7 “TUNEL staining of 128-cell stage embryos injected with miR-4561 mimics, inhibitor, and NC.” Please add a description to the callout about the red arrow in the figure.

Response 8:

Thank you for your question. We have added the description to the callout about the red arrow in the Figure 7E.

Question 9:

Line 330: Figure 8 The schematic “Inducing embryo apoptosis” should revised to “Induce embryo apoptosis”.

Response 9:

We have revised “Inducing embryo apoptosis” to “Induce embryo apoptosis”. Due to the addition of some experimental data, Figure 8 has been adjusted to Figure 9.

Question 10:

Line 335: Animal feeding and *V. anguillarum* challenge section, what's the water temperature during the adult healthy lampreys temporarily raised? Why choose the *V. anguillarum* bacteria to challenge the lampreys? What are the symptoms of *V. anguillarum* infection?

Response 10:

Thank you for your question. The living environment of lampreys is more complicated because of the migration, and it is difficult to clarify the specific environmental temperature. But after many years of exploration, at 20°C, the breeding conditions of lampreys are stable and the immune effect is quite good.

V. anguillarum has been detected in the seawater where we fished for lamprey, and a small amount of *Vibrio* bacteria has also been identified in the lampreys (please refer to the reference 1 below). Our previous studies have proved that *V. anguillarum* can trigger a variety of immune responses in lampreys (please refer to the reference 2-6). Another reason we prefer to use this bacteria is that it can be used as a double verification of Gram-negative bacteria (LPS) stimulation.

1. Li Y, Xie W, Li Q. Characterisation of the bacterial community structures in the intestine of *Lampetra morii*. *Antonie Van Leeuwenhoek*. 2016 Jul;109(7):979-86. doi: 10.1007/s10482-016-0699-0. Epub 2016 May 14. PMID: 27180095.

2. Teng H, Wang D, Lu J, Zhou Y, Pang Y, Li Q. Novel insights into the evolution of the caveolin superfamily and mechanisms of antiapoptotic effects and cell proliferation in lamprey. *Dev Comp Immunol*. 2019 Jun;95:118-128. doi: 10.1016/j.dci.2019.01.005. Epub 2019 Feb 8. PMID: 30742851.

3. Wang D, Gou M, Hou J, Pang Y, Li Q. The role of serpin protein on the natural immune defense against pathogen infection in *Lampetra japonica*. *Fish Shellfish Immunol*. 2019 Sep;92:196-208. doi: 10.1016/j.fsi.2019.05.062. Epub 2019 Jun 5. PMID: 31176010.

4. Lv W, Ma A, Chi X, Li Q, Pang Y, Su P. A novel complement factor I involving in the complement system immune response from *Lampetra morii*. *Fish Shellfish Immunol*. 2020 Mar;98:988-994. doi: 10.1016/j.fsi.2019.11.017. Epub 2019 Nov 8. PMID: 31712129.

5. Zhu Y, Li J, Li Q, Pang Y. Characterization of lamprey (*Lampetra japonica*) tnfr10-like gene: A potential granulocyte marker molecule and its immune functions. *Mol Immunol*. 2020 Aug;124:25-34. doi: 10.1016/j.molimm.2020.05.015. Epub 2020 Jun 1. PMID: 32497752.

6. Han Q, Han Y, Wen H, Pang Y, Li Q. Molecular Evolution of Apolipoprotein Multigene Family and the Original Functional Properties of Serum Apolipoprotein (LAL2) in *Lampetra japonica*. *Front Immunol.* 2020 Aug 11;11:1751. doi: 10.3389/fimmu.2020.01751. PMID: 32849624; PMCID: PMC7431520.

Question 11:

Line 340: “A total of 0.1 mL of *V. anguillarum* (1×10^7 CFU/mL) and LPS (0.1 mg/mL) (Sigma Aldrich, St. Louis, Missouri, USA) were individually injected into the abdomen of healthy lampreys (three for each group).” Please add here the basis for the concentration of the bacterial infection or pre-experimental description.

Response 11:

Regarding the injection concentration of *V. anguillarum* and LPS, please refer to the following two papers. As you can see, in this article and in our previous papers, we have used a concentration of 1×10^7 CFU/mL, which can already produce an immune response. In previous experiments, we also tried the concentration of 1×10^8 CFU/mL with a stronger immune response, but it would cause a certain number of fish deaths. For the purpose of reducing animal deaths, we set the stimulation concentration to the 1×10^7 CFU/mL.

V. anguillarum

1. Wang D, Gou M, Hou J, Pang Y, Li Q. The role of serpin protein on the natural immune defense against pathogen infection in *Lampetra japonica*. *Fish Shellfish Immunol.* 2019 Sep;92:196-208. doi: 10.1016/j.fsi.2019.05.062. Epub 2019 Jun 5.

LPS

2. Pang Y, Xiao R, Liu X, Li Q. Identification and characterization of the lamprey high-mobility group box 1 gene. *PLoS One.* 2012;7(4):e35755. doi: 10.1371/journal.pone.0035755.

Line 418: “Supraneural body tissues were dissected from lamprey and digested with 0.04% collagenase I (Sigma Aldrich, USA) for 10 min at 18 °C to collect 1×10^6 cells/mL primary cells.” Should revised to “Supraneural body tissues were dissected from lamprey and digested with 0.04% collagenase I (Sigma Aldrich, USA) for 10 min at 18 °C to collect 1×10^6 cells/mL primary cells.”

Response 12:

We have revised “Supraneural body tissues were dissected from lamprey and digested with 0.04% collagenase I (Sigma Aldrich, USA) for 10 min at 18 °C to

collect 1×10^6 cells/mL primary cells.” to Supraneural body tissues were dissected from lamprey and digested with 0.04% collagenase I (Sigma Aldrich, USA) for 10 min at 18 °C to collect 1×10^6 cells/mL primary cells. Please refer to Page 19, Line 437-438.

Question 13:

Line 428: “Supraneural body cells and HEK293 cells were transfected with 30–200 nM of each oligonucleotide.”, confirm the HEK293T or HEK293.

Response 13:

We confirm the cell line is HEK293.

Question 14:

Line 472-473: “p-values were considered significant and are indicated with asterisks in the figures: * $P < 0.05$, ** $P < 0.01$, *** $P < 0.001$.”, confirm the “ *** $P < 0.01$, ** $P < 0.001$ ”, they are same now.

Response 14:

We are so sorry for our carelessness. We have corrected this error in the revised manuscript, Please refer to Page 21, Line 503.

Reviewer #3 (Remarks to the Author):

This article investigated whether microRNAs play a key regulatory role in the immunity of lamprey. The authors studied the changes of microRNAs expression in leukocytes of lamprey following *Vibrio anguillarum* infection. Using comparative methods, they identified some microRNAs potentially involved in immune regulation in lamprey. Among these microRNAs, they discovered that lamprey miR-4561 targets LIP mRNA and downregulates the LIP expression. miR-4561 not only participates in the lamprey innate immune response, but also affects embryonic development.

This paper showed lamprey microRNAs may significantly affect cellular immunity and apoptosis by regulating gene expression. This is the first study to investigate microRNAs implicated in the antibacterial defense of lamprey. This work is meaningful and valuable for further research on the functions of microRNAs in jawless vertebrates. However, there are some shortcomings:

1. While the authors observed that overexpression of miR-4561 induced apoptosis

in embryonic cells, inhibition of miR-4561 insignificantly affect embryonic development. Additionally, the authors did not examine whether the expression of miR-4561 changes during embryonic development. Thus, the authors need to conduct further experiments to gather more evidences to confirm that miR-4561 is indeed involved in embryonic development under physiological conditions.

Response 1:

Thank you for your question, we have added relevant experimental data to clarify this problem, so we have added Figure 8 to the revised manuscript. The details are as follows, you can also check the Page 12-14, line 253-279 of the revised manuscript.

In order to further confirm that miRNA4561 targeting *lip* causes apoptosis in lamprey embryo development, three siRNAs (siRNA-LIP308, siRNA-LIP515, siRNA-LIP942) were designed to silence lamprey *lip* gene expression. Fig. 8A show the fertilization rate after microinjection of lamprey and prove that the data can be used for subsequent experiment. Fig. 8B represents a survival rate showing the neurula stage and head stage after microinjection of siRNA, respectively, silencing *lip* gene expression significantly died after neurula stage, and the silencing effect of siRNA308 was higher. The embryo development map of lamprey after siRNA interference in Fig. 8C. TNF activates NF- κ B mainly by TNFR pathways, whereas TNF binding to TNFR can induce apoptosis (18). We consequently predicted that the *lip* gene loss of expression may play a role in the signal transmission pathway of persistent apoptosis in individuals. We through RT-qPCR to verify the 10 gene in LIP - TNF - NF- κ B signal axis, 10 pairs of primers were designed and the same RNA aliquots were assayed in triplicate (Fig. 8D). The results showed that *lip* gene silencing could induce apoptosis through activate TNF and NF- κ B signaling pathways. At the same time, we found that miR-4561 induces the activate TNF and NF- κ B signaling pathways and trigger apoptosis by targeting LIP (Fig. 8E). miR-4561 targeted *lip* gene loss of expression in lamprey results in apoptosis, cell growth arrest and immune response, all of which serve to growth, development and immune processes in lamprey.

Figure 8 miR-4561 targeted LIP induces apoptosis through activate TNF and NF-κB signaling pathways. (A) Fertilization rate of siRNA silenced lamprey *lip* gene expression. (B) Survival rate of siRNA silenced lamprey *lip* gene expression in gastrulation embryos. (C) Developmental map of siRNA silenced lamprey *lip* gene expression. After silencing the *lip* gene, lampreys are all lethal after neurula stage and head stage. (D) Changes of apoptosis-related genes in lamprey neurula stage after silencing *lip* gene by siRNA. (E) Changes of apoptosis-related genes in lamprey neurula stage after miRNA4561 targeting *lip* gene.

2. There are some mistakes in this article. The descriptions of Figure 3 are

inaccurate in line 151-153. The authors described “either a miR-4561 mimic or inhibitor” in line 175, but only a miR-4561 mimic was used in Figure 4B. “The expression of miR-4561 was normalized by U6 RNA” should be deleted in line 177.

In conclusion, this work is meaningful but the details of the work needs some improvements.

Response 2:

Thank you for your question.

The descriptions of Figure 3 are inaccurate in line 151-153.

We are really sorry for our carelessness. We used the wrong version of the picture before, which caused the legend and the figure to be inconsistent. We have fixed this problem now, please refer to Page 6, Line 148-152.

The authors described “either a miR-4561 mimic or inhibitor” in line 175, but only a miR-4561 mimic was used in Figure 4B.

We are really sorry for our carelessness. We have corrected it in the revised manuscript, please refer to Page 7, Line 174.

“The expression of miR-4561 was normalized by U6 RNA” should be deleted in line 177.

We have deleted this sentence in our revised manuscript.

Reviewers' comments:

Reviewer #1 (Remarks to the Author):

It looks like that the manuscript is sufficient for publication.

Reviewer #2 (Remarks to the Author):

In this study, the authors found the microRNA targeting LIP, an important immune protein of lampreys, and preliminarily explored the role of miR-4561 in the immune response caused by Gram-negative bacteria of lampreys. It is the first study to investigate microRNAs implicated in the antibacterial defense of lamprey. Based on the new manuscript, most errors have been modified. But I still found some errors in this manuscript, so I give my recommendation for minor revision. My specific comments are as follows:

Question 1:

The lampreys (*Lampetra morii*) used in this manuscript live in fresh water for life, not in sea water, please check the correctness in the response of "Besides, *V. anguillarum* has been detected in the seawater where we fished for lamprey." (from 7086_1_rebuttal_231334_qs97lg) and "Adult healthy lampreys (*Lampetra morii*) (weight, 35 ± 5 g) were collected from the Yalu River and temporarily raised in a culture system composed of an aquarium (100 × 50 × 50 cm), black ceramsite sand filtration system (2–3 mm³), and ceramsite sand bottom layer (12 ± 2 cm high). Water height was about 40 cm and the water flow was about 2 L/min." in this manuscript. In the sentence, "Water height...", is the water here sea water or fresh water?

In the reference 1. Li Y, Xie W, Li Q. Characterisation of the bacterial community structures in the intestine of *Lampetra morii*. *Antonie Van Leeuwenhoek*. 2016 Jul;109 (7):979-86. doi: 10.1007/s10482-016-0699-0. Epub 2016 May 14. PMID: 27180095, there no details for the *Vibrio anguillarum*, just only the genus of vibrio. And the lampreys come from Songhua River: "Wild lampreys were obtained from the Tongjiang Valley, a branch of the Songhua River in Heilongjiang Province in China. These lampreys were kept in Fiber Reinforced Plastic (FRP) tanks with fresh water at 4–16 °C before the experiments."

In the section of "A total of 0.1 mL of *V. anguillarum* (1×10^7 CFU/mL) and LPS (0.1 mg/mL)...", what is the number of strain *V. anguillarum* ? where it is isolated from? how to culture the bacteria? And add the reference of bacteria's concentration of 1×10^7 CFU/mL and details explain.

Question 2:

Figure 3: Please add significance analysis to the annotation of Figure 3(E). Similarly, please unify it in Figure 5-8.

Question 3:

Figure 6: Please add corresponding data of untreated group in Figure 6(A) and Figure 6(B).

Question 4:

Line 462: The basis for the use of concentrations of *V. anguillarum* infection needs to be clarified in this section.

Question 5:

Line 489: "Three groups of ,100 fertilized embryos were injected with lip siRNA (Shanghai GenePharma Co.,Ltd RNAi duplex with sense sequence: 5' CCGCAACCGUGAGUUCUJUTT3' and antisense sequence: 5' AAAGAACUCACGGUUGCGTT3')" should be revised to "Three groups of 100 fertilized embryos were injected with lip siRNA (Shanghai GenePharma Co.,Ltd RNAi duplex with sense sequence: 5' CCGCAACCGUGAGUUCUJUTT3' and antisense sequence: 5' AAAGAACUCACGGUUGCGTT3')".

Reviewer #3 (Remarks to the Author):

The authors have improved their manuscript. Their revisions strengthen the conclusion. I think it is acceptable for publication.

Reviewer #2 (Remarks to the Author):

In this study, the authors found the microRNA targeting LIP, an important immune protein of lampreys, and preliminarily explored the role of miR-4561 in the immune response caused by Gram-negative bacteria of lampreys. It is the first study to investigate microRNAs implicated in the antibacterial defense of lamprey. Based on the new manuscript, most errors have been modified. But I still found some errors in this manuscript, so I give my recommendation for minor revision.

My specific comments are as follows:

Question 1:

In this study, the authors found the microRNA targeting LIP, an important immune protein of lampreys, and preliminarily explored the role of miR-4561 in the immune response caused by Gram-negative bacteria of lampreys. It is the first study to investigate microRNAs implicated in the antibacterial defense of lamprey. Based on the new manuscript, most errors have been modified. But I still found some errors in this manuscript, so I give my recommendation for minor revision.

My specific comments are as follows:

Question 1:

The lampreys (*Lampetra morii*) used in this manuscript live in fresh water for life, not in sea water, please check the correctness in the response of “Besides, *V. anguillarum* has been detected in the seawater where we fished for lamprey.” (from 7086_1_rebuttal_231334_qs97lg) and “Adult healthy lampreys (*Lampetra morii*) (weight, 35 ± 5 g) were collected from the Yalu River and temporarily raised in a culture system composed of an aquarium ($100 \times 50 \times 50$ cm), black ceramsite sand filtration system ($2\text{--}3$ mm³), and ceramsite sand bottom layer (12 ± 2 cm high). Water height was about 40 cm and the water flow was about 2 L/min.” in this manuscript. In the sentence, “Water height...”, is the water here sea water or fresh water?

In the reference 1. Li Y, Xie W, Li Q. Characterisation of the bacterial community structures in the intestine of *Lampetra morii*. *Antonie Van Leeuwenhoek*. 2016 Jul;109 (7):979-86. doi: 10.1007/s10482-016-0699-0. Epub 2016 May 14. PMID:

27180095, there no details for the *Vibrio anguillarum*, just only the genus of vibrio. And the lampreys come from Songhua River: “Wild lampreys were obtained from the Tongjiang Valley, a branch of the Songhua River in Heilongjiang Province in China. These lampreys were kept in Fiber Reinforced Plastic (FRP) tanks with fresh water at 4–16 °C before the experiments.”

In the section of “A total of 0.1 mL of *V. anguillarum* (1×10^7 CFU/mL) and LPS (0.1 mg/mL)...”, what is the number of strain *V. anguillarum* ? where it is isolated from? how to culture the bacteria? And add the reference of bacteria’s concentration of 1×10^7 CFU/mL and details explain.

Response 1:

We are so sorry for this mistake and we confirm the water here is fresh water. There are three types of lampreys in China, *Lampetra morii* and *Lampetra reissneri* live in fresh water, and *Lampetra japonica* is a migratory species in the river and sea.

In addition, we very appreciate you for your question. We had confirmed the genus and species names of *Vibrio anguillarum* through Institute of microbiology, Chinese Academy of Sciences. We used *Vibrio anguillarum* as a representative of Gram-negative bacteria, which has been demonstrated a strong immune response in several articles of lampreys.

Each lamprey was immunized with 100 μ L *V.anguillarum* (suspended to 1×10^7 cells/mL in normal saline), the number of strain *V. anguillarum* is 1×10^6 cells in each lamprey. *Vibrio anguillarum* strains were isolated from the intestine of the lamprey. The *V. anguillarum* (28°C) strain was cultured in 2216E liquid medium with 0.5% peptone, 0.1% yeast extract (pH=8.0). Moreover, we use the bacteria’s concentration of 1×10^6 cells ~ 1×10^7 cells in each lamprey according to the weigh of lampreys. Please refer to the following references. We also added this part to the materials and methods, please refer to Page 17, Line 374-379.

References:

1. Hou J, **Pang Y**, Li Q. Comprehensive Evolutionary Analysis of Lamprey TNFR-Associated Factors (TRAFs) and Receptor-Interacting Protein Kinase (RIPKs) and Insights Into the Functional Characterization of TRAF3/6 and RIPK1. *Front Immunol.* 2020;11:663.
2. Han Q, Han Y, Wen H, **Pang Y**, Li Q. Molecular Evolution of Apolipoprotein Multigene Family and the Original Functional Properties of Serum Apolipoprotein (LAL2) in *Lampetra japonica*. *Front Immunol.* 2020;11:1751.
3. Li C, Wang D, Guan X, Liu S, Su P, Li Q, **Pang Y**. HMGB1 from *Lampetra*

japonica promotes inflammatory activation in supraneural body cells. Dev Comp Immunol. 2019;92:50-59.

Question 2:

Figure 3: Please add significance analysis to the annotation of Figure 3(E). Similarly, please unify it in Figure 5-8.

Response 2:

Thank you for your question. We have added significance analysis to the annotation of Figure 3(E) and Figure 5-8. Please check the figure legends of Figure 3 and Figure 5-8.

Question 3:

Figure 6: Please add corresponding data of untreated group in Figure 6(A) and Figure 6(B).

Response 3:

Thank you for your question. We have added the data untreated group in Figure 6. Please refer to Page 9-10, Line 216-220.

Question 4:

Line 462: The basis for the use of concentrations of *V. anguillarum* infection needs to be clarified in this section.

Response 4:

Thank you for your question. We have added the basis for the use of concentrations of *V. anguillarum* infection and relevant references. Please refer to Page 20, Line 470-471 and the reference 42.

Question 5:

Line 489: “Three groups of 100 fertilized embryos were injected with lip siRNA (Shanghai GenePharma Co.,Ltd RNAi duplex with sense sequence: 5’ CCGCAACCGUGAGUUCUUUTT3’ and antisense sequence: 5’ AAAGAACUCACGGUUGCGGTT3’)” should be revised to “Three groups of 100 fertilized embryos were injected with lip siRNA (Shanghai GenePharma Co.,Ltd RNAi duplex with sense sequence: 5’ CCGCAACCGUGAGUUCUUUTT3’ and antisense sequence: 5’ AAAGAACUCACGGUUGCGGTT3’)”.

Response 5:

Thank you for your question. We have revised “Three groups of 100 fertilized embryos were injected with lip siRNA (Shanghai GenePharma Co.,Ltd RNAi duplex with sense sequence: 5’ CCGCAACCGUGAGUUCUUUTT3’ and antisense

sequence: 5' AAAGAACUCACGGUUGCGGTT3')” to “Three groups of 100 fertilized embryos were injected with lip siRNA (Shanghai GenePharma Co.,Ltd RNAi duplex with sense sequence: 5' CCGCAACCGUGAGUUCUUUTT3' and antisense sequence: 5' AAAGAACUCACGGUUGCGGTT3')”. Please refer to Page 20-21, Line 496-499.

REVIEWERS' COMMENTS:

Reviewer #2 (Remarks to the Author):

In this study, the authors found the microRNA targeting LIP, an important immune protein of lampreys, and preliminarily explored the role of miR-4561 in the immune response caused by Gram-negative bacteria of lampreys. The authors have improved their manuscript. Their revisions strengthen the conclusion. Based on the new revision, I have found some errors in this manuscript, I think it is acceptable for publication after minor revision.

My specific comments are as follows:

Question 1:

Line 145: "This is consistent with the role of LIP in the response to Gram-negative bacteria and pathogen clearance." The results in this part do not seem to indicate "pathogen clearance", please modify.

Question 2:

Line 312: "Recently, it has been reported that the sea anemone, poreforming toxin sticholysin II plays an important role in eukaryotic cell lysis." Please briefly describe the relationship between poreforming toxin sticholysin II and LIP, more fully associated with the functional domain of them.

Question 3:

Line 329: "miiuy croaker" should be revised to "miiuy croaker (*Miichthys miiuy*)".

Question 4:

Line 421: "Membranes were then incubated with HRP-conjugated anti-rabbit IgG and anti-mouse IgG antibodies (Abcam, Cambridge, MA, USA) used at 1:5000 dilution." Please show the final concentration of the antibody, consistent with the above.

Reviewer #2 (Remarks to the Author):

In this study, the authors found the microRNA targeting LIP, an important immune protein of lampreys, and preliminarily explored the role of miR-4561 in the immune response caused by Gram-negative bacteria of lampreys. The authors have improved their manuscript. Their revisions strengthen the conclusion. Based on the new revision, I have found some errors in this manuscript, I think it is acceptable for publication after minor revision.

My specific comments are as follows:

Question 1:

Line 145: “This is consistent with the role of LIP in the response to Gram-negative bacteria and pathogen clearance.” The results in this part do not seem to indicate “pathogen clearance”, please modify.

Response 1:

Thank you for your question. We have deleted "pathogen clearance " to make the sentence more accurate. Please refer to Page 5, Line 134-135.

Question 2:

Line 312: “Recently, it has been reported that the sea anemone, poreforming toxin sticholysin II plays an important role in eukaryotic cell lysis.” Please briefly describe the relationship between poreforming toxin sticholysin II and LIP, more fully associated with the functional domain of them.

Response 2:

Thank you for your question. We have added a brief description of the relationship between poreforming toxin sticholysin II and LIP and added related reference in our revised manuscript. Please refer to Page 8, Line 256-261 and reference 27.

Question 3:

Line 329: “miiuy croaker” should be revised to “miiuy croaker (*Miichthys miiuy*)”.

Response3:

We have revised “miiuy croaker” to “miiuy croaker (*Miichthys miiuy*) ”. Please refer to Page 9, Line 275.

Question 4:

Line 421: “Membranes were then incubated with HRP-conjugated anti-rabbit IgG and anti-mouse IgG antibodies (Abcam, Cambridge, MA, USA) used at 1:5000 dilution.” Please show the final concentration of the antibody, consistent with the above.

Response 4:

Thank you for your question. We have added the final concentration of the antibody in our revised manuscript. Please refer to Page 11, Line 363-365.